# Asymmetric localization of DLC1 defines avian trunk neural crest polarity for directional delamination and migration

Jessica Aijia Liu[1], Yanxia Rao[1], May Pui Lai Cheung[1], Man-Ning Hui[2], Ming-Hoi Wu[1], Lo-Kong Chan[3], Irene Oi-Lin Ng[3], Ben Niu[1], Kathryn S.E. Cheah[1], Rakesh Sharma[4], Louis Hodgson [5] & Martin Cheung[1]

Following epithelial-mesenchymal transition, acquisition of avian trunk neural crest cell (NCC) polarity is prerequisite for directional delamination and migration, which in turn is essential for peripheral nervous system development. However, how this cell polarization is established and regulated remains unknown. Here we demonstrate that, using the RHOA biosensor in vivo and in vitro, the initiation of NCC polarization is accompanied by highly activated RHOA in the cytoplasm at the cell rear and its fluctuating activity at the front edge. This differential RHOA activity determines polarized NC morphology and motility, and is regulated by the asymmetrically localized RhoGAP Deleted in liver cancer (DLC1) in the cytoplasm at the cell front. Importantly, the association of DLC1 with NEDD9 is crucial for its asymmetric localization and differential RHOA activity. Moreover, NC specifiers, SOX9 and SOX10, regulate *NEDD9* and *DLC1* expression, respectively. These results present a SOX9/SOX10-NEDD9/DLC1-RHOA regulatory axis to govern NCC migratory polarization.

[1] School of Biomedical Sciences, Li Ka Shing Faculty of Medicine, The University of Hong Kong, Hong Kong, China. [2] Department of Obstetrics and Gynaecology, Li Ka Shing Faculty of Medicine, The University of Hong Kong, Hong Kong, China. [3] State Key Laboratory for Liver Research and Department of Pathology, Li Ka Shing Faculty of Medicine, The University of Hong Kong, Hong Kong, China. [4] Proteomics and Metabolomics Core Facility, Li Ka Shing Faculty of Medicine, The University of Hong Kong, Hong Kong, China. [5] Department of Anatomy and Structural Biology, Gruss-Lipper Biophotonics Center, Albert Einstein College of Medicine, Bronx, NY 10461, USA. Jessica Aijia Liu and Martin Cheung contributed equally to this work. Correspondence and requests for materials should be addressed to M.C. (email: mcheung9@hku.hk)

Delamination and migration of neural crest cells (NCCs) is one of the defining embryonic events in vertebrate development and dysregulation of these processes leads to various human diseases[1]. Multipotent NCCs originate in the dorsal neural tube and implement a transcriptional programme to initiate an epithelial-mesenchymal transition (EMT) that involves alteration of cytoskeletal structure, loss of cell-cell adhesion and apical-basal polarity to convert an epithelial cell into a mesenchymal motile phenotype[2]. At the onset of delamination, NCCs acquire front-back polarity that is essential for directional migration into the periphery where they differentiate into neurons and glia[3]. However, characteristic molecular circuits regulating NCC morphodynamics at this stage are not known. The RHO family of small GTPases acts as molecular switches, cycling between an active GTP-bound and an inactive GDP-bound state. RHO GTPases play key roles in regulating cytoskeletal dynamics, polarization and adhesion during cell migration[4–6]. Previous studies demonstrated that functional requirements for the levels of RHO signaling in EMT differ between species or cranial vs. trunk NCCs[7–11]. However, conflicting studies using distinct RHO inhibitors treatment at the same axial level have not clearly defined whether RHO plays a positive or negative role in trunk NC delamination[8–10]. These differing reports likely reflect the fact that RHO signaling is highly context dependent where precise spatiotemporal dynamics of RHO activity determines cellular phenotypes. Previous studies have shown that localized RHO activation in a discrete apical region of premigratory cranial NCCs is essential for detachment from the neuroepithelium[12]. So far, there is not yet a clear picture of RHO activity in emigrating trunk NCCs in favor of a specific GTPase at subcellular scales. Moreover, how this dynamic cartography of the RHO GTPase activity is spatially regulated and coordinated with NC polarity and migratory behavior are unknown. We thus examined the regulatory principles that link activity of RHO signaling and their regulators to trunk NCC polarization and directional migration.

In this study, using a fluorescence resonance energy transfer (FRET)-based biosensor for RHOA in chick embryos and live-cell imaging in vitro, we demonstrate that high RHOA activity in the cytoplasm defines the prospective rear of delaminating and migratory trunk NCCs while fluctuating RHOA activity in the membrane protrusion at the cell front. This differential RHOA activity is regulated by asymmetric localization of the RhoGAP Deleted in liver cancer 1 (DLC1) in the cytoplasm at the cell front. Gain- and loss-of-function studies reveal that appropriate level of DLC1 activity is essential for the establishment of NC polarity through spatial restriction of RHOA activity at the back, which is prerequisite for directional delamination and migration. Importantly, this differential distribution of DLC1 depends on its binding partner, NEDD9 and their association is required for the establishment of polarized rather than the total level of RHOA activity. Last, DLC1 and NEDD9 genes are subject to transcriptional regulation by NC specifiers SOX10 and SOX9 respectively. Together, these results reveal a SOX9/SOX10-NEDD9/DLC1-RHOA regulatory axis to orient trunk NCCs in the direction of movement.

## Results

**Asymmetric active RHOA indicates NC migratory polarization**. Although subcellular localization of RHOA activity has been shown to be crucial in determining polarity of different cell types[13], whether RHOA also engages in establishing trunk NC polarity is largely unknown. To investigate the spatiotemporal dynamics of RHOA signaling in delaminating and migrating NCCs, we electroporated caudal neural tube of chick embryos in Hamburger and Hamilton stages[14] (st) 11–12 with a RHOA single chain FRET-based biosensor that was previously used to detect localized RHOA activation in fibroblasts[15]. At 24 h post-transfection (hpt), electroporated embryos at st 15–16 were harvested for FRET analysis on cross-section of the thoracic neural tube where NC delamination and migration are ongoing. FRET imaging and measurement of the FRET index between the back and front of NCCs revealed high RHOA activity in the cytoplasm of the cell rear relative to low or fluctuating RHOA activity in membrane protrusions at the leading front of delaminating/early- and late-migrating NCCs, which are marked by NC specifiers, SOX9 and SOX10, respectively (Fig. 1a–d). In contrast, moderate level of RHOA activity is detected and uniformly distributed throughout neuroepithelial cells in the neural tube (Supplementary Fig. 1a, b). These results indicate that NCCs display differential RHOA activity in subcellular localization as they undergo directional delamination and migration. To examine the dynamics of RHOA activity in live NCCs, we electroporated neural tube explants with FRET biosensor and performed time-lapse imaging of NCCs undergoing directional delamination onto fibronectin-coated dishes. Consistent with in vivo observations, imaging results revealed that RHOA activity was highly enriched at the cell rear and also dynamically localized in membrane protrusions at the leading edge of polarized NCCs emigrating from the explants (Fig. 1e and Supplementary Movie 1). To determine the persistence of RHOA activity, we measured the FRET index between the back and front of polarized NCCs over time[4]. We found that RHOA activity was persistently high at the back while fluctuating levels were observed at the front during directional migration over time (Fig. 1f). Analysis of the total FRET signal showed that RHOA activity was higher at the back than to the front (Fig. 1g). Clustering of the FRET index ratio between the back and front over a 20 min of live cell imaging confirmed the maintenance of polarized RHOA activity during migration (Fig. 1h). To further interrogate whether this differential RHOA activity is maintained as NCCs change in migratory direction, emigrating NCCs from neural tube explants were exposed to beads coated with stromal cell derived factor 1 (SDF-1), which is a chemoattractant for trunk NCCs[16], to mimic the in vivo environment. We observed initial polarization of RHOA activity along the front-back axis and the subsequent re-localization of high RHOA to the prospective cell rear was synchronized with the establishment of new membrane protrusions at the cell front as NCCs underwent directional change in response to SDF-1 (Fig. 1i and j and Supplementary Movie 2). Quantification of the FRET index between the back and front over time revealed the maintenance of high RHOA activity at the cell rear even when NCCs changed their direction of movement (Fig. 1k). Thus, pre-existing asymmetrical localization of RHOA activity defines the cell's eventual direction of polarization. To further correlate asymmetric RHOA activity with cell polarization, we examined RHOA dynamics in a population of emigrating NCCs, which undergo front-back switch in response to SDF-1. Time-lapse FRET imaging showed that NCC with a front-back polarized morphology gradually acquired elevated RHOA activity at the front (A) which eventually became the back of the cell following cell repolarization together with the formation of a new membrane protrusion at the front pointing toward SDF-1 (B) (Fig. 1l and m and Supplementary Movie 3). Quantification of the FRET activity between A and B over time revealed that redistribution of differential RHOA activity preceded initiation of a gradual front and back switch (Fig. 1n). Altogether, in vitro neural tube explant studies further consolidate in vivo observations that existing asymmetry of RHOA activity sometimes

indicates the future back-front polarity and directional migration of trunk NCCs.

**DLC1 is asymmetrically expressed in migratory NCCs.** In search for genes involved in regulating polarized activity of RHOA, we identified RhoGTPase-activating protein (RhoGAP),

Deleted in liver cancer 1 (DLC1), which has been shown to play a key role in the inhibition of tumor growth and metastasis through suppression of RHO signaling[17, 18]. The chick has three major transcriptional isoforms of the *DLC1* gene (Fig. 2a). Comparative sequence analysis revealed that their amino acid sequences exhibit extensive overall homology with their counterparts in both human and mouse, with the greatest

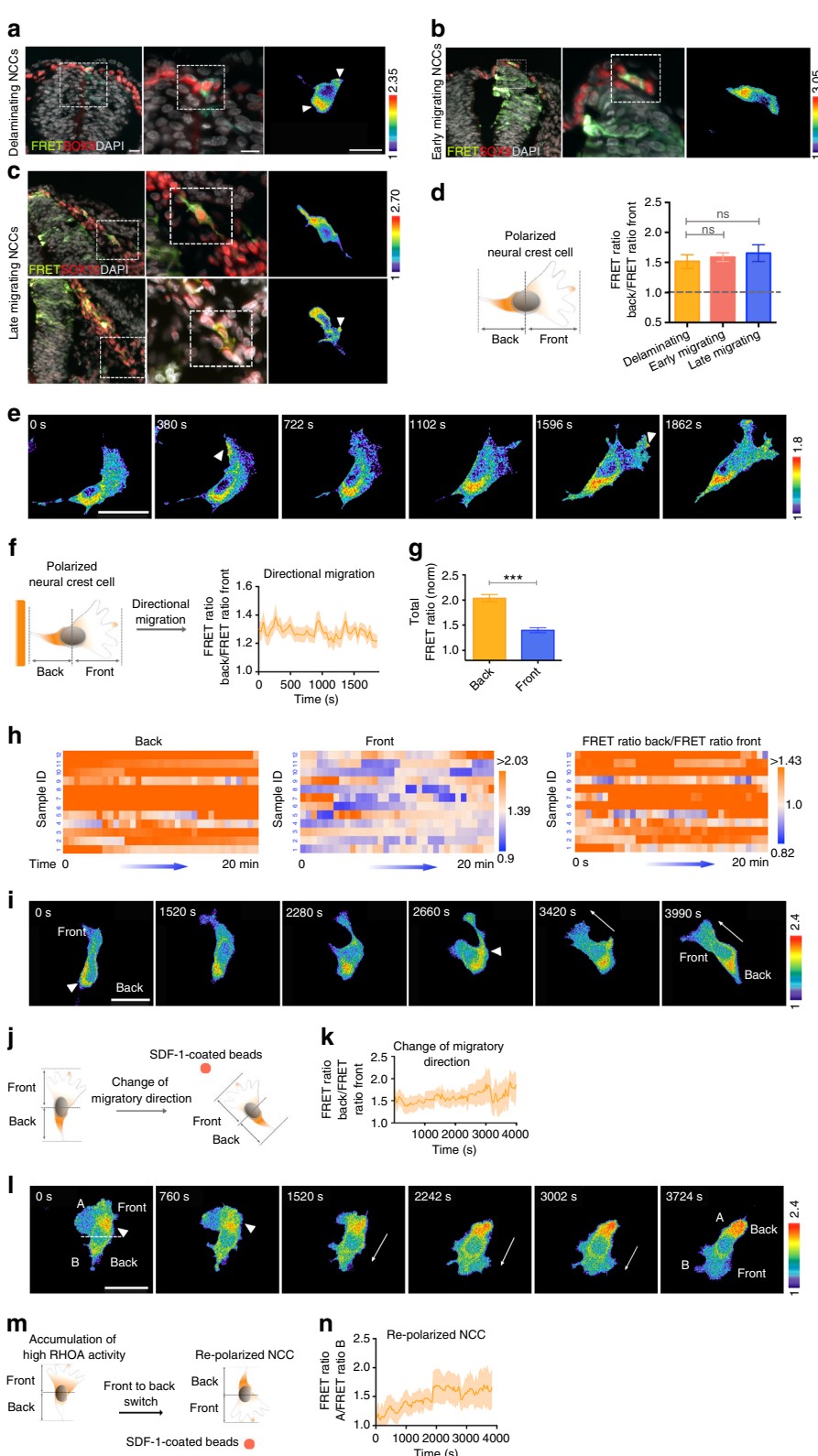

similarities in the four functional domains, the sterile α motif domain (SAM), the focal adhesion targeting (FAT) region, the RhoGAP and the steroidogenic acute regulatory domain (START) (Supplementary Fig. 2a, b). We first examined *DLC1* expression from st 8 to 14 chick embryos using a riboprobe against the conserved RhoGAP domain of the three isoforms (Fig. 2a). *DLC1* mRNA was initially detected at st 8 and 9 in the premigratory and delaminating cranial NCCs that co-express with NC specifier, SOX9, and the migratory NC marker, HNK-1 (Fig. 2d, e). *DLC1* expression was maintained in migratory cranial NCCs in the midbrain, hindbrain regions and in the frontonasal process from st 11–14 (Fig. 2d). In the posterior trunk from st 11 to 13, *DLC1* exhibited a similar expression pattern to SOX9 and other NC specifiers, SOX10 and SNAIL2, in early migrating NCCs (Fig. 2d, f). Coinciding with the rostral to caudal gradient of NC development, *DLC1* was first initiated in the premigratory and maintained in the migratory cranial NC population followed by expression in early migratory trunk NCCs. To further delineate which isoforms were expressed in the trunk NCCs, we took advantage of a *SOX10* enhancer (*SOX10*-E1) to drive EGFP reporter expression specifically in vagal/trunk NCCs[19]. This allowed us to enrich this population of cells by fluorescence activated cell sorting (FACS) for qPCR analysis with isoform-specific primers (Fig. 2b). The results showed that *DLC1 isoform 3* was highly expressed in sorted EGFP⁺ cells compared to low levels expression of *DLC1 isoforms 1* and *2* (Fig. 2c), indicating that *isoform 3* is predominantly expressed in trunk NCCs. For simplicity, we therefore named isoform 3 as *DLC1* for the rest of the analyses.

Strikingly, immunofluorescence analysis of neural tube explants showed that DLC1 exhibited asymmetric localization in the cytoplasm at the front of emigrating and migrating NCCs (Fig. 2g–j) but was not colocalized with the focal adhesion kinase (FAK) as shown in previous studies[20] (Fig. 2k). Similarly, when V5-tagged DLC1 was ectopically expressed at low level in chick neural tube, we observed preferential cytoplasmic localization of DLC1 to the front of delaminating and early migrating NCCs (Supplementary Fig. 3a). To quantify asymmetric expression of DLC1, we measured signal intensity by line scan analysis along the cell body from different protrusions at the leading edge (Fig. 2h). The results revealed that DLC1 expression was enriched at the cell front in the cytoplasm between membrane protrusions and nucleus, whereas expression was low in the nucleus and at the back, indicating that DLC1 exhibited polarized expression at the onset of NC delamination (Fig. 2l). We then compared co-localization patterns between RHOA activity and DLC1 fluorescence intensity from the leading cell edge to perinuclear region of cytoplasm in different groups of cells (Fig. 2m), and confirmed a negative correlation with high DLC1 expression in

the region of low RHOA activity and vice versa (Fig. 2n, o), suggesting that DLC1 may be involved in establishing polarized RHOA activity in delaminating NCCs.

**DLC1 regulates polarized RHOA activity and NC migration.** To further investigate whether DLC1 regulates spatial restriction of RHOA activity for the establishment of NC polarity and directional migration, we overexpressed FRET biosensor together with DLC1, which harbors a strong RhoGAP activity to inhibit active form of RHOA-GTP expression as determined by RHOA pull-down activation assay, into the caudal hemineural tube of st 10–11 chick embryos (Fig. 3a, b). At 24hpt, analysis of FRET signals on sections showed that RHOA activity was reduced or barely detectable in DLC1 overexpressing NCCs, which exhibited lack of front-back polarity axis (Fig. 3c). In addition, overexpression of DLC1 did not induce ectopic expression of SOX9, SOX10 and HNK-1 in the neuroepithelium, suggesting that DLC1 is not sufficient to trigger NC formation. However, expression of these markers in migratory NCCs was reduced on the transfected side (Fig. 3e–h and u). Strikingly, neuroepithelial cells overexpressing DLC1 were observed delaminating not only from the basal surface of the dorsal neural tube where laminin was lost but also into the lumen (Fig. 3i, j). Consistently, N-Cadherin (N-Cad) expression was lost at the apical surface of dorsal neuroepithelium (Fig. 3k, l), indicating that overexpression of DLC1 disrupted apical-basal polarity of dorsal neural tube cells. These data suggest that high level of DLC1 expression reduced RHOA activity, resulting in aberrant polarity of premigratory NCCs that delaminate into the neural tube lumen instead of following their normal migratory route.

To examine whether inhibition of DLC1 function affected RHOA activity, NCC fate, polarity and migratory behavior, we generated a dominant negative version of DLC1 (DN-DLC1), consisting of an amino-terminal region of DLC1 without the RhoGAP and START domains (Fig. 3a). This construct was previously shown to impair DLC1 association with its partner factor, resulting in inhibition of cell motility in vitro[21]. By contrast to DLC1, overexpression of DN-DLC1 resulted in increased level of RHOA-GTP (Fig. 3b). To further demonstrate a direct inhibitory effect of DN-DLC1 on DLC1, we co-electroporated DN-DLC1 with DLC1 and found the level of RHOA-GTP was slightly reduced compared to the control and DLC1 (Fig. 3b), indicating that DN-DLC1 indeed functions as a dominant negative to inhibit endogenous RhoGAP activity of DLC1. Consistently, overexpression of DN-DLC1 in vivo resulted in a marked elevation of RHOA activity, which was evenly distributed throughout the cytoplasm of delaminating NCCs without distinct subcellular localization and cells appeared round

---

**Fig. 1** Asymmetric RHOA activity correlates with NC polarity and directed migration. Processed FRET signal images of delaminating **a**, early migrating **b** and late migrating NCCs **c**. The magnified area and selected cells are marked with dotted squares. White arrowheads indicate FRET signal. Scale bars, 50 µm. **d** Schematic representation of asymmetric RHOA activity in a polarized NC. Quantification of the ratio of the FRET index between the back and front of delaminating ($n = 107$), early migrating ($n = 123$) and late migrating NCCs ($n = 196$). 25 embryos were analyzed. Mean ± s.e.m. Student's t-test, ns, not significant. **e** Processed FRET signal images of time-lapse series of delaminating NCCs. White arrowheads indicate RHOA activity at the cell front. Scale bar, 50 µm. **f** Schematic representation of asymmetric RHOA activity in a polarized NC undergoing directional migration. Quantification of the ratio of the FRET index between the back and front over time. $n = 81$ cells from 26 explants were analyzed. **g** Quantification of the total FRET index in the indicated region. Mean±s.e.m.; $n = 81$; Student's $t$ test; ***$p < 0.0001$. **h** Heat maps representing the FRET index as a function of time at the back or at the front, and the FRET ratio between the back and front of 12 selected NCCs from 12 explants. **i** Processed FRET signal images of time-lapse series of a migrating NC. White arrows indicate the change of migratory direction toward SDF-1. Scale bar, 50 µm. **j** Schematic representation of RHOA activity distribution in a polarized NC undergoing directional change of movement toward SDF-1 coated beads. **k** Quantification of the ratio of the FRET index between the back and front of migratory NCCs. $n = 81$ cells from 21 explants were analyzed. **l** Example of a stationary NC undergoing front and back switch in response to SDF-1. $n = 72$ from 17 explants were analyzed. White arrowheads indicate initial accumulation of high RHOA activity in the cell front. White arrows indicate direction of cell repolarization. Scale bar, 50 µm. **m** Schematic representation of the example shown in Fig. 1l. **n** Quantification of the ratio of the FRET index between A and B of the example shown in Fig. 1l. Mean ± s.e.m

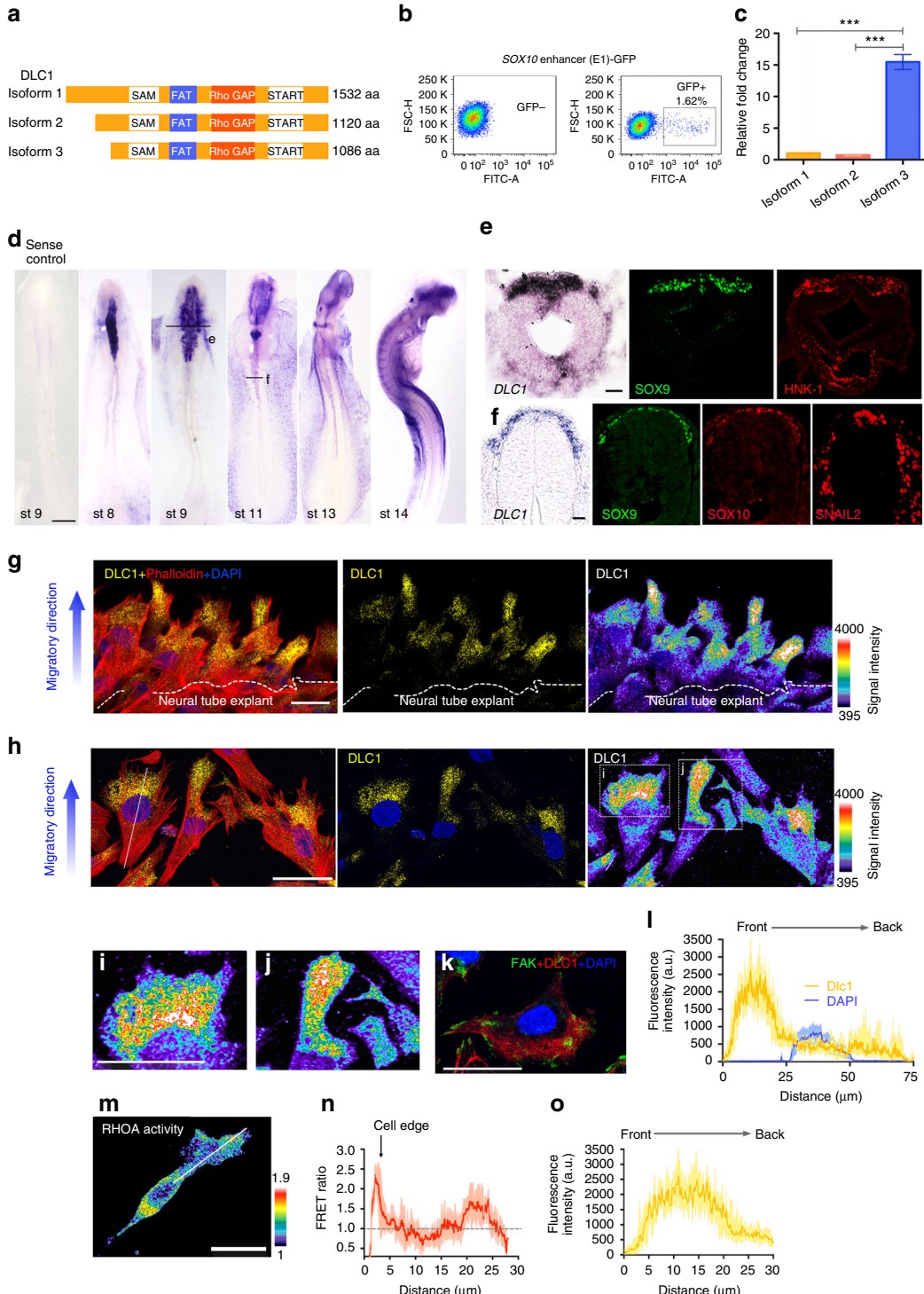

**Fig. 2** Asymmetric localization of DLC1 negatively correlates with RHOA activity. **a** Schematic representation of three chick DLC1 isoforms harboring four conserved regions: the sterile α motif (SAM) domain, the focal adhesion targeting (FAT) domain, the Rho GTPase activating (RhoGAP) domain and the steroidogenic acute regulatory (START) domain. **b** Flow cytometry enriched trunk NCCs (~ 1.62%) labeled by *SOX10* enhancer (E1)-driven GFP expression. **c** mRNA expression levels of *DLC1* isoforms in sorted NCCs. Mean ± s.e.m. Student's *t*-test, ***$p < 0.0001$ **d** Sense riboprobe for *DLC1* serves as a negative control. In situ hybridization of *DLC1* from stages (st) 8–14 chicken embryos. Scale bar, 150 μm. **e** Cross section at the cranial region of a st 9 chick embryo stained with *DLC1* (black line) and immunofluorescence for SOX9 and HNK-1 on consecutive sections. **f** Cross section at the trunk region of a st 11 chick embryo stained with *DLC1* (black line) and immunofluorescence for SOX9, SOX10 and SNAIL2 on consecutive sections. Scale bars, 50μm.

Immunofluorescence for DLC1 and phalloidin on delaminating **g** and early migratory NCCs **h** from neural tube explants and nuclei are stained with DAPI. DLC1 channels are also shown in pseudocolor. White dotted lines outline the border of neural plate explant. **i, j** Magnification of the boxed regions with enriched DLC1 expression shown in pseudocolor. **k** Immunofluorescence for FAK and DLC1 on migratory NCCs and nuclei are stained with DAPI. **l** Line scans analysis showing the average fluorescence intensity of DLC1 along the white dotted line in **h**. $n = 47$. **m** Representative image for the measurement of RHOA activity along the white line. Scale bars, 20μm. **n** Quantification of RHOA activity ($n = 26$) and **o** DLC1 fluorescence intensity ($n = 47$) from the leading cell front to the perinuclear region of cytoplasm

in shape (Fig. 3c). Quantitative analysis of total FRET activity in vivo revealed that, compared to vector control, the overall levels of RHOA activation were significantly increased (Student's $t$-test, $p < 0.0001$) and reduced (Student's $t$-test, $p < 0.001$) in NCCs overexpressing DN-DLC1 and DLC1, respectively (Fig. 3d). Analysis of NC markers revealed that overexpression of DN-DLC1 resulted in markedly fewer

migratory NCCs expressing HNK-1 compared to the contralateral side (Fig. 3m–p and u). Consistently, the amount of early migratory NCCs expressing SOX9 and SOX10 were reduced on the transfected side but their expression in the premigratory domain remains unaltered, indicating that NC identity was not affected by overexpression of DN-DLC1 (Fig. 3q, r). In contrast to DLC1, the majority of NCCs expressing DN-DLC1 remained in

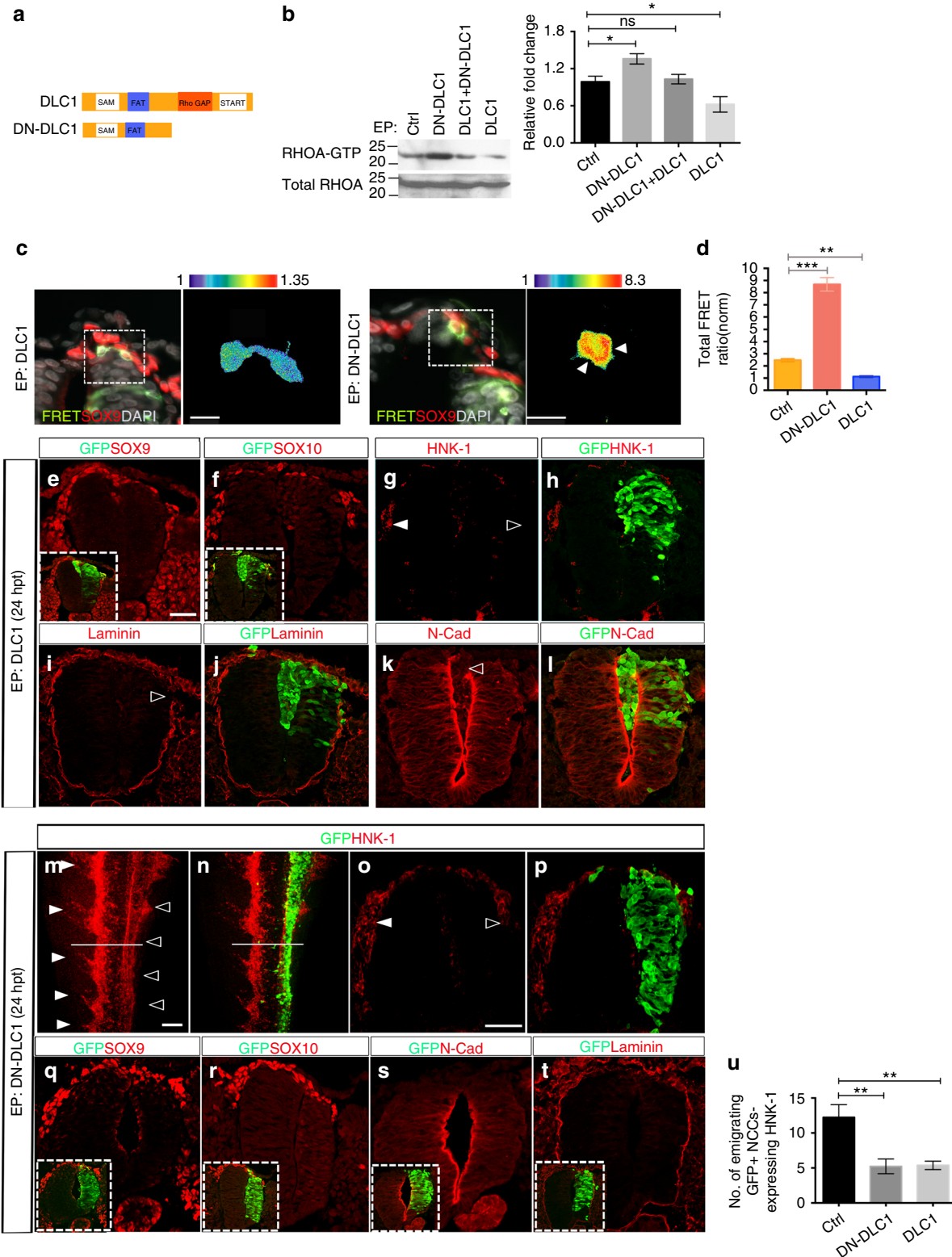

the neuroepithelium without disrupting apical-basal polarity (Fig. 3s, t). To further demonstrate that DLC1 is required for NC polarity and delamination, we electroporated a fluorescein-tagged morpholino targeting DLC1 isoform 3 (DLC1-MO) into the caudal hemineural tube of st 10–11 embryos. At 24 hpt, DLC1 protein was diminished using higher dosage of DLC1-MO, whereas its expression remained unaltered in embryos treated with control-MO (Ctrl-MO) and vector alone (Supplementary Fig. 4a). Similar reduction of DLC1 expression was observed in DLC1-MO-treated cultured NCCs, which appeared round in shape without distinct front-back polarity like DN-DLC1 expressing cells (Supplementary Fig. 4b and Fig. 3c). In agreement with this, cells expressing DLC1-MO resulted in reduced expression of $FOXD3^+$ migrating NCCs compared to the untransfected side and the Ctrl-MO-treated embryos (Supplementary Fig. 4c). These results demonstrate functional requirements for DLC1 in the establishment of NC polarity and delamination. Altogether these in vivo studies indicate that appropriate level of DLC1 activity is required for the spatial restriction of RHOA activity, the establishment of NC polarity and directional delamination of NCCs.

**DLC1 spatially restricts RHOA for directed NC migration.** To further evaluate how alteration of DLC1 activity affects the dynamics of NC migratory behavior in more details, we performed in vitro time-lapse imaging of delaminating NCCs from the neural tube explants electroporated with vector control, DLC1 and DN-DLC1 and examined the effects of cell polarity based on the degree of membrane protrusions[22, 23]. Since the formation of membrane protrusions is driven by actin dynamics, NCCs were also transfected with Lifeact-mCherry to label actin cytoskeleton for quantification of the angles between membrane protrusion and migratory direction (Fig. 4a). As expected, GFP expressing cells migrated out of the explants, and exhibited a characteristic pattern of membrane protrusions at the cell front toward the direction of migration (Fig. 4a and Supplementary Movie 4). By contrast, NCCs overexpressing DLC1 lost front-back polarization with randomly oriented and erratic finger-like projections and exhibited loss of directionality (Fig. 4a and Supplementary Movie 5). Membrane protrusions in the form of blebbing were observed in DN-DLC1 expressing cells, which exhibited limited motility (Fig. 4a and Supplementary Movie 6). We then measured a set of migration parameters to determine how alteration of NC polarity by manipulation of DLC1 activity affected migratory behavior. The travel distance, persistence and net speed of cell migration were reduced in both DLC1 and DN-DLC1 overexpressing NCCs compared to the vector

control and non-GFP expressing cells (Fig. 4b–e), indicating lack of directionality. The total speed of migrating NCCs expressing DN-DLC1 was markedly lower than that of DLC1 overexpressing cells, which migrated at a comparable speed to the control and non-GFP expressing cells (Fig. 4f). These results indicate that overexpression and dominant-negative inhibition of DLC1 activity disrupted NCC polarization and directional migration. We then evaluated their effects on RHOA activity in cultured NCCs transfected with FRET-biosensor in the presence of SDF-1 to determine whether polarized morphology and migratory behavior of NCCs could be restored by a guidance signal (Fig. 4g). Consistent with in vivo observations, time-lapse imaging and the FRET index between the back and front revealed that overexpression of DLC1 and DN-DLC1 resulted in marked reduction and elevation of RHOA activity throughout the cell respectively compared to the vector control, which exhibited a reproducible polarized distribution of RHOA activity with persistent high FRET signal at the back and migrated toward SDF-1 (Fig. 4h, i and Supplementary Movies 7–9). In addition, we observed a similar lack of cell polarity and directional movement in both treatments, suggesting that addition of SDF-1 in neural tube explant culture was not able to restore the morphological and migration defects caused by aberrant RHOA signaling (Fig. 4h). Altogether, these in vitro studies further consolidate in vivo results that delaminating NCCs require precise spatial activity of DLC1 for restricting RHOA activity high at the back that is essential for the acquisition of polarized NCC morphology and motility toward the source of chemoattractant.

**NEDD9 is required for the asymmetric localization of DLC1.** Previous studies showed that DLC1 interacts with other factors to modulate its RhoGAP function[17]. To identify protein factors associated with DLC1, we performed immunoprecipitation with DLC1 antibody specific for isoform 3 using chick neural tube protein lysates. A proteomic analysis revealed different functional categories of proteins enriched by DLC1 (Fig. 5a, Supplementary Fig. 5a, b and Supplementary Data 1). The Ingenuity Pathway Knowledge base analysis showed that the majority of these proteins were involved in EIF2 signaling, remodeling of epithelial adherens junctions, and regulation of actin-based motility by RHO (Fig. 5b and Supplementary Data 2). Consistently, interactome network indicated that proteins related to cell motility and cytoskeleton regulation were highly associated with DLC1 (Fig. 5c). NEDD9, a member of the Cas family of scaffolding proteins, was chosen for further study because it has been shown to be involved in NC motility[24] although the

**Fig. 3** DLC1 spatially restricts RHOA activity and regulates NC migratory behavior. **a** Schematic diagram of a full-length DLC1 and the dominant-negative DLC1 (DN-DLC1) containing the C-terminally truncated fragment of DLC1 without the RhoGAP and START domains. **b** Immunoblot for RHOA-GTP on protein lysates extracted from neural tubes electroporated with the indicated constructs at 24 hpt. Total RHOA is used as a loading control. Bars represent results from densitometric analysis (mean ± s.e.m., $n = 3$ independent experiments). Student's $t$-test, *$p < 0.05$; ns, not significant. **c** Processed FRET signal images of cross-sections from embryos electroporated with the FRET probe plus DLC1 or DN-DLC1. NCCs are marked by SOX9 immunofluorescence and nuclei are stained with DAPI. The magnified area and selected cells are marked with dotted squares. White arrowheads indicate FRET signal. Scale bars, 20 μm. **d** Quantification of the total FRET index in control ($n = 425/25$ embryos), NCCs expressing DN-DLC1 ($n = 148/21$ embryos) and DLC1 ($n = 131/16$ embryos). Student's $t$-test, **$p < 0.001$; ***$p < 0.0001$. **e–l** Immunofluorescence for SOX9, SOX10, HNK-1, Laminin and N-Cad on transverse sections of embryos electroporated (EP) with DLC1 at 24 hpt ($n = 12$). Insets show the merge images of GFP and endogenous SOX9 or SOX10 expression. White triangle indicates endogenous HNK-1 expression in migratory NCCs, whereas open triangles indicate reduced expression of HNK-1, Laminin and N-Cad on the transfected side of the neural tube. Scale bars, 50μm. **m, n** Immunofluorescence for HNK-1 in an embryo electroporated with DN-DLC1 at 24 hpt ($n = 17$). Scale bar, 150μm. **o, p** Cross-section at the level of the white line in m and n. White triangles indicate endogenous HNK-1 expression in migratory NCCs, whereas open triangles indicate reduced HNK-1 expression in the transfected side. **q–t** Immunofluorescence for SOX9, SOX10, N-Cad and Laminin on transverse sections of embryos electroporated with DN-DLC1 at 24 hpt ($n = 17$). Insets show merge images of green and red channels. Scale bar, 50μm. **u** Quantification of the number of emigrating GFP$^+$ cells expressing HNK-1 in the transfected side of the neural tube electroporated with the indicated constructs. Average of cells counted from at least 30 sections of 10 embryos per treatment. Mean ± s.e.m. Student's $t$-test **$p < 0.001$

mechanism of action is not known. Co-immunoprecipitation confirmed an association of endogenous DLC1 and NEDD9 proteins in the embryonic neural tubes (Fig. 5d), raising the possibility that DN-DLC1 may interfere with the DLC1-NEDD9 interaction based on its mode of action from a previous study[21].

Indeed, ectopic expression of DN-DLC1 was also able to co-immunoprecipitate with NEDD9. Further domain mapping revealed that the C-terminally truncated version of DN-DLC1 comprising of SAM and FAT domains [DN-DLC1 (320)][25] where most interaction partners of DLC1 bind could associate with

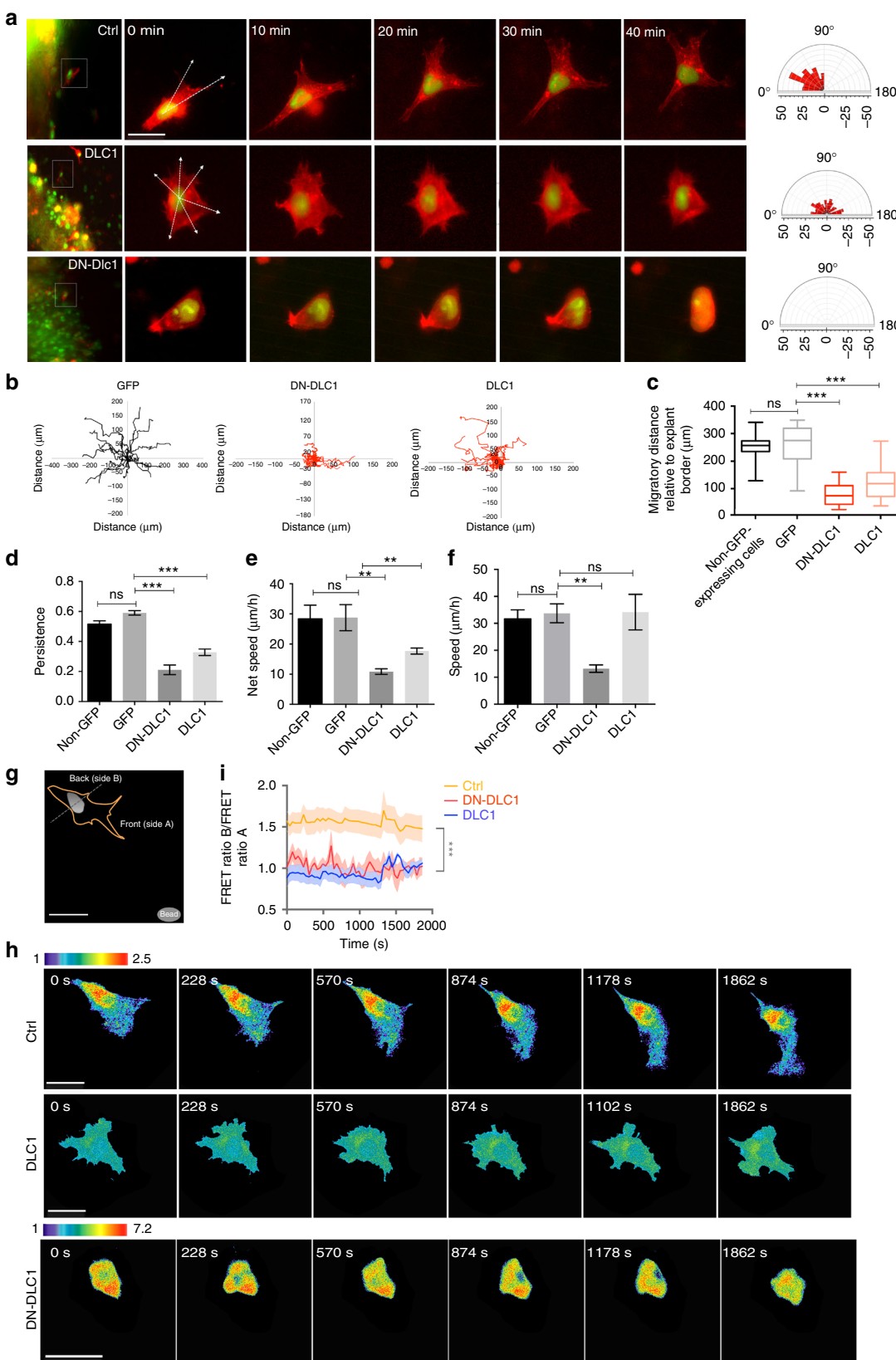

NEDD9, whereas SAM domain alone did not show interaction with NEDD9 (Fig. 5e). Consistently, overexpression of DN-DLC1 or DN-DLC1 (320) but not the SAM domain alone was able to reduce NC delamination (Supplementary Fig. 6a). These results suggest that NEDD9 could be sequestered from its endogenous interaction with DLC1 through binding to the FAT domain of DN-DLC1.

In situ hybridization on consecutive sections show that *NEDD9* mRNA was detected in the premigratory and early migratory trunk NCCs of st 11 embryos in a similar manner to *DLC1*, *SOX9* and *SOX10*. In addition, *NEDD9* expression was also notable in the floor plate. The distribution of *NEDD9* transcripts resemble its protein expression which was detected in the cytoplasm of the premigratory and emigrating NCCs in an overlapping manner with nuclear localization of SOX9 and SOX10 proteins, indicating co-localization of these factors at the onset of NC delamination (Fig. 5f). Immunofluorescence analysis indicated that NEDD9 exhibited a similar pattern to DLC1 with polarized localization between the leading edge and the nucleus of emigrating NCCs (Fig. 5g). We then examined whether the spatial expression of DLC1 is altered in cells treated with NEDD9 morpholino (NEDD9-MO), which reduced levels of NEDD9 protein in delaminating NCCs, compared to the untransfected side and embryos treated with control-MO (Ctrl-MO) (Supplementary Fig. 7a). NEDD9-MO resulted in aberrant and elongated morphology of NCCs without discernible front-back polarity (Fig. 5h). Importantly DLC1 protein exhibited random distribution throughout the NEDD9-MO transfected cells as shown by line scan analysis while polarized DLC1 expression remained unaltered in Ctrl-MO-treated cells (Fig. 5h, i), suggesting that the asymmetric localization of DLC1 is regulated by association with its binding partner NEDD9.

**Association of DLC1 with NEDD9 restricts active RhoA**. Since NEDD9 has been implicated in regulating Rho GTPases in cancer cells[26], the above findings raise two possibilities. The loss of NEDD9 function might lead to aberrant RHOA activity resulting in loss of cell polarity. Alternatively, mislocalization of DLC1 is the primary cause for the dysregulated NCC polarity in NEDD9-MO. To distinguish these possibilities, we first examined RHOA activity in delaminating NCCs expressing NEDD9-MO in vivo and in neural tube explants treated with SDF-1 (Fig. 6a, c). Surprisingly quantification of the total FRET activity in cells treated with NEDD9-MO in vivo was comparable to the vector control and NEDD9 overexpressing cells (Fig. 6b), suggesting that loss of NEDD9 function and increased level of NEDD9 expression have no direct impact on total RHOA activity. However, we observed mislocalization of RHOA activity around the nucleus of NEDD9-MO-treated cell, which exhibited aberrant front-back polarization and lack of directionality in migration (Fig. 6a, c and Supplementary Movie 10). Consistently the FRET index between the back and front of NEDD9-MO-treated cells did not exhibit differential character compared to the control and NEDD9

overexpressing cells (Fig. 6b–d, f). These data suggest that lack of RHOA polarity in the absence of NEDD9 function could be attributed to the dysregulated localization of DLC1, which disrupted precise spatial RHOA restriction without altering the total level of RHOA activity. This prompted us to further examine whether association of DLC1 with NEDD9 is functionally required for the establishment of RHOA polarity. We then overexpressed DN-DLC1 to impair interaction of endogenous DLC1 with NEDD9, rendering DLC1 incapable for spatial restriction of RHOA activity at the cell rear. If that was the case, polarized RHOA activity could be restored by overexpression of NEDD9 in DN-DLC1 expressing cells. Indeed, the differential distribution of RHOA activity between the back and front in DN-DLC1 + NEDD9 expressing cells in vivo was restored with the FRET index comparable to the control and NEDD9 overexpression alone, but its total FRET signal remained higher than other treatments (Fig. 6a, b). Clustering of the FRET index between the back and front of cultured NCCs expressing DN-DLC1 + NEDD9 also showed restoration of differential RHOA activity like NEDD9 overexpressing cells, and cells exhibited polarized morphology and directional migration (Fig. 6c, e, f and Supplementary Movies 11 and 12). In agreement with this, restoration of NC delamination was observed in embryos electroporated with DN-DLC1 + NEDD9 as opposed to a marked reduction of SOX10[+] and HNK-1[+] cells in the transfected side of embryos electroporated with DN-DLC1 or NEDD9-MO, whereas overexpression of NEDD9 alone did not affect NC delamination (Fig. 6g, h). Collectively, these data show that interaction of DLC1 with NEDD9 is essential for the spatial restriction of RHOA activity at the back of the cell and the leading edge that is prerequisite for the establishment of a front-rear polarity axis and directional delamination and migration of trunk NCCs.

**SOXE factors regulate *DLC1* and *NEDD9* genes expression**. The SOXE transcription factors SOX9 and SOX10 play important roles in regulating trunk NC specification and delamination[27, 28]. The overlapping expression of *DLC1*, *NEDD9*, *SOX9* and *SOX10* in premigratory and early migratory NCCs prompted us to examine whether *DLC1* and *NEDD9* genes expression are subjected to the transcriptional regulation by SOX9 and/or SOX10. The results revealed that SOX9 or SOX10 was sufficient to induce ectopic expression of *DLC1* throughout the neural tube at 12–24 hpt but not 6 hpt (Fig. 7a and Supplementary Fig. 8a, b). Considering SOX9 induced *SOX10* expression[29], *DLC1* was not induced in the neural tube electroporated with SOX9 plus SOX10-MO at 24 hpt (Supplementary Fig. 8c). Consistently, downregulation of *DLC1* expression was observed in SOX10-MO treated embryos, compared to Ctrl-MO (Fig. 7b). These results show that SOX9 induces SOX10 to regulate *DLC1* expression. To further investigate whether DLC1 could mediate SOX9 and SOX10-induced NC delamination and migration, we electroporated SOX9 or SOX10 plus DN-DLC1 and examined their

---

**Fig. 4** Appropriate level of DLC1 activity is required for directional migration of NCCs. **a** Time-lapse imaging showing the protrusion dynamics of emigrating NCCs expressing GFP control, DLC1 and DN-DLC1. White arrows indicate protrusions direction. Polar histogram plots showing distribution of the angle between actin protrusions and subsequent migration direction from all the time series. $n = 22/10$ explants for GFP control, $n = 20/12$ explants for DLC1, $n = 25/12$ explants for DN-DLC1. Scale bar, 20 μm. **b** Migration tracks of 17 representative NCCs expressing GFP control, DN-DLC1 and DLC1 are shown. Quantification of the total migratory distance **c**, persistence **d**, net speed **e** and total speed **f** of NCCs electroporated with the indicated constructs. GFP and non-GFP cells serve as controls. Mean±s.e.m. Student's *t*-test, ***$p < 0.0001$; **$p < 0.001$; ns, not significant. **g** Schematic diagram showing the back (side B) and front side (side A) of a neural crest cell migrating toward SDF-1 coated beads. Scale bar, 50 μm. **h** Processed FRET signal images of time-lapse series of emigrating NCCs from neural tube explants treated with vector control ($n = 111/16$ explants), DLC1 ($n = 93/17$ explants) and DN-DLC1 ($n = 88/13$ explants). Scale bars, 50 μm. **i** Quantification of the ratio of the FRET index between the back (B) and front (A) of the cluster for the indicated conditions. Mean±s.e.m. Student's *t*-test, ***$p < 0.0001$

impact on NC motility using *FOXD3* as a migratory NC marker (Fig. 7c, d). While overexpression of SOX9 or SOX10 promoted robust NC delamination and migration, electroporation of SOX9 + DN-DLC1 resulted in almost complete abrogation of NC delamination and migration (Fig. 7c, d). The effects were relatively mild in embryos electroporated with SOX10 + DN-DLC1 with reduced *FOXD3*⁺ migrating NCCs compared to the

untransfected side (Fig. 7d). Together, these data demonstrate that SOX10 promoted NC migration through regulation of *DLC1*. We then examined whether SOX9 and/or SOX10 regulates *NEDD9* expression. Electroporation of SOX9 resulted in ectopic expression of *NEDD9* at 6 and 12 hpt throughout the dorsal-ventral axis of the neural tube compared to non-electroporated side (Fig. 7e and Supplementary Fig. 8d). In addition, SOX9

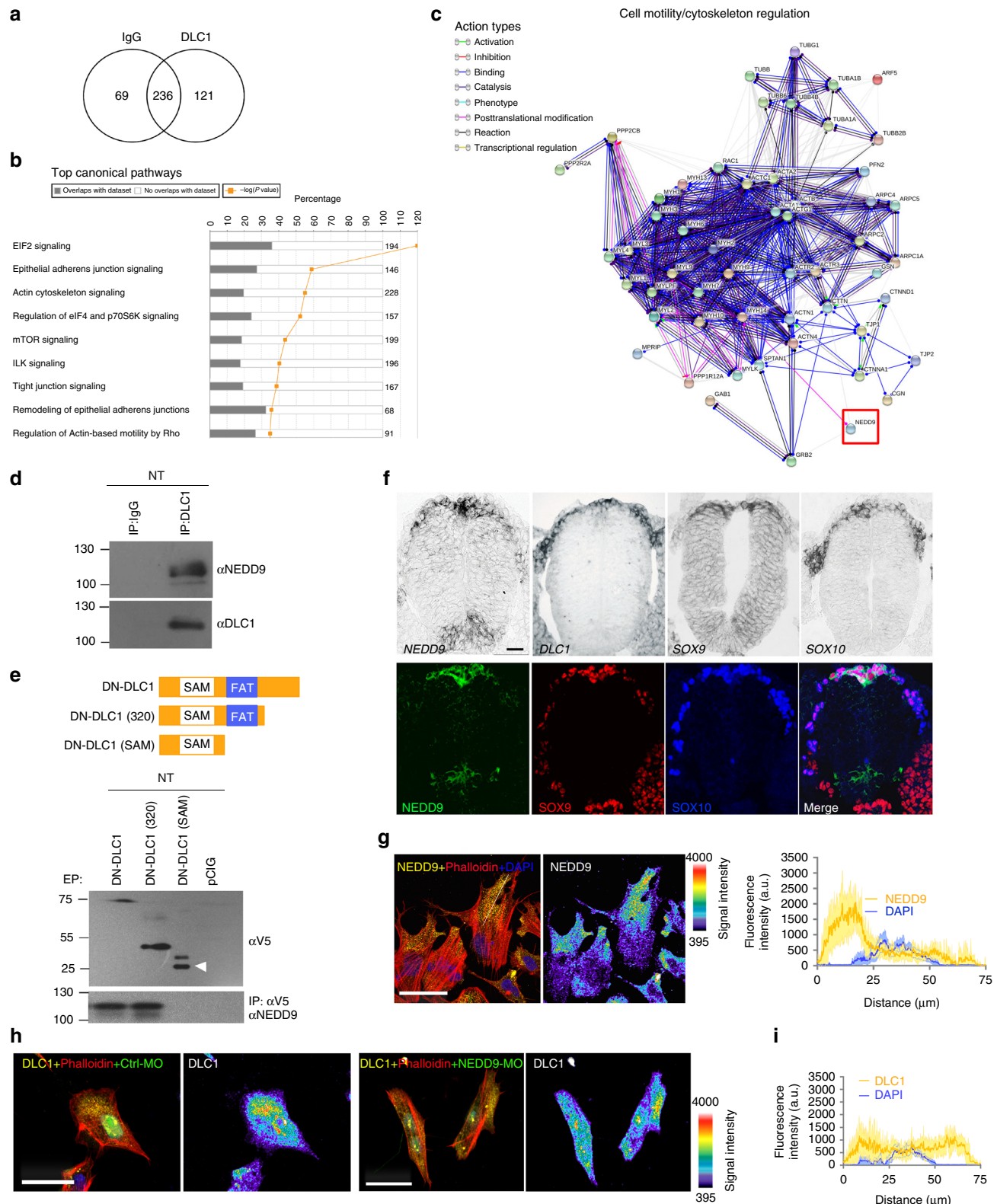

expressing cells were co-localized with ectopic NEDD9 protein expression, indicating that SOX9 induces *NEDD9* expression in a cell-autonomous manner (Supplementary Fig. 8d). In agreement with this, depletion of SOX9 expression in SOX9-MO treated embryos resulted in downregulation of *NEDD9* at 24 hpt compared to Ctrl-MO (Fig. 7f), indicating that SOX9 is required for *NEDD9* expression. By contrast, depletion of SOX10 protein expression by Sox10-MO did not affect *SOX9* and *NEDD9* expression levels, and NEDD9 expression was still induced in SOX9 + SOX10-MO expressing cells (Supplementary Fig. 8e, f), suggesting that SOX10 is not required for *NEDD9* expression. Consistently, overexpression of SOX10 was not able to induce ectopic NEDD9 expression though it triggered HNK-1 expression at 12 hpt (Supplementary Fig. 8g). Chromatin immunoprecipitation and reporter assays showed that *NEDD9* is a direct transcriptional target of SOX9 but not SOX10 through binding to *NEDD9* enhancer sequences but not promoter region (Fig. 7g, h and Supplementary Fig. 9a–d). Thus, these results indicate a specific role for SOX9 in the direct transcriptional regulation of *NEDD9* expression. Accordingly, epistasis analysis revealed that cells expressing SOX9 + NEDD9-MO remained in the neuroepithelium and the number of HNK-1$^+$ migratory NCCs on the transfected side was reduced compared to the vector control and embryos electroporated with SOX9 or SOX9 + Ctrl-MO (Fig. 7j, k). Taken together, these results show that SOX9 promotes NC delamination and migration by direct transcriptional activation of *NEDD9* expression.

## Discussion

During EMT, premigratory NCCs lost apical-basal polarity to delaminate from the dorsal neural tube followed by acquisition of a front-rear polarity axis that is pre-requisite for directional migration to their correct destinations for differentiation[30, 31]. While several studies have revealed the molecular control for regulating the cellular events during cranial NC EMT and directional migration[7, 12, 32–34], candidate factors and the underlying mechanisms for trunk NCCs to acquire polarization and directional migration are not yet well understood. Our study reveals a SOXE-DLC1/NEDD9-RHOA regulatory axis to establish avian trunk NCC polarity for directional delamination and migration (Fig. 8).

Small Rho GTPases spatiotemporally coordinate cell polarization and migration in variety types of cell[4–6, 15]. Previous studies in trunk NCCs utilized the Rho-binding domain of Rhotekin for the detection of active GTP-bound forms of RHO proteins in the membrane[10]. In addition, functional studies have largely relied on the use of distinct Rho inhibitors with differential potencies[7, 8] or on expression of dominant-negative[11]. Although these studies have demonstrated the involvement of RHO signaling in NC motility, these approaches are potentially non-specific targeting

RHOA, RHOB and RHOC activities or their downstream effectors resulting in either promoting or inhibiting NC delamination[8–10]. These conflicting findings reflect the fact that precise subcellular localization and activity of Rho GTPases control different cellular events, which cannot be precisely controlled in loss-of-function experiments. Moreover, which specific Rho GTPases are involved in establishing NC polarity remains elusive. Our study for the first time reveals a unique spatiotemporal profile of RHOA GTPase activity in delaminating and migrating trunk NCCs using FRET biosensor in chick embryos and neural tube explants. We provide two major insights in this study. First, we found that the majority of RHOA active GTP-bound form is localized in the cytoplasm at the cell back and also dynamically expressed at the leading edge that coincides with acquisition of a back-to-front polarity axis aligned with the prospective vector of migration. Second, the persistence of high RHOA activity in cytoplasm correlates with the eventual cell rear as NCCs undergo re-polarization concomitant with the change in migratory direction in response to chemoattractant, SDF-1. Thus, the asymmetric localization of RHOA activity is a key molecular signal that mediates cell orientation to the direction of movement. In contrast, previous FRET analysis in *Xenopus* embryos revealed that RHOA activity is localized to the cell periphery and elevated at the site of collision of two cranial NCCs leading to contact inhibition of locomotion[32]. NCCs arising from different axial levels may account for the differences in the subcellular localization of RHOA activity but its level of activity is similarly required for directional migration of both cranial and trunk NCCs. While cranial NCCs adopt contact inhibition of locomotion to govern their collective migratory behavior, trunk NCCs migrate as single cell chains and interact extensively with neighboring cells[35]. A recent report demonstrated that leader cells are crucial for orchestrating the orderly movement of trunk NCCs through cell-cell contact with follower cells[36]. Based on these studies together with our results, it is tempting to speculate that leader cells may provide guidance signals through dynamic interaction with the followers to maintain differential RHOA activity and front-back polarity axis for directed migratory patterns of trunk NCCs. Further analysis will be required to address this issue.

The cytoplasmic localization at the rear-cell body suggest that RHOA might reside in the endoplasmic reticulum or endosomal vesicles as described previously[37], serving as reservoir of active RHOA to be translocated to the leading edge of migrating cells. In addition, this pool of active RHOA has also been shown to regulate cellular contractility via microtubule depolymerization[38], while dynamic activation of RHOA at the leading edge controls actin fiber assembly and focal adhesion dynamics to coordinate both extension and retraction of membrane protrusions at the cellular front[39]. The mechanisms of RHOA activation in both sub-cellular regions of delaminating NCCs are largely unknown.

**Fig. 5** NEDD9 as DLC1 interacting protein is required for the polarized expression of DLC1. **a** Venn diagram showing the number of proteins enriched by DLC1. **b** DLC1-associated proteins involved in major canonical pathways as assessed by the Ingenuity Pathway Knowledge base. **c** Interactome network of DLC1 and its associated proteins involved in cell motility and cytoskeleton regulation. **d** Immunoprecipitation (IP) was performed in neural tube lysate using anti-DLC1, and IgG as control. Immunoprecipitated proteins were resolved on sodium dodecyl sulfate-polyacrylamide gel electrophoresis gels, and Western blotting was performed with anti-NEDD9 and anti-DLC1 antibodies. **e** Schematic showing the design of the C-terminally truncated DN-DLC1 constructs. IP with anti-V5 was performed in lysates extracted from neural tubes electroporated with the indicated constructs followed by western blot with anti-V5 or anti-NEDD9. White arrowhead indicates the specific protein band of DN-DLC1 (SAM). **f** In situ hybridization for *NEDD9, DLC1, SOX9* and *SOX10* on transverse sections of st 11 chick embryo. Immunofluorescence for NEDD9, SOX9 and SOX10 on transverse sections of st 11 chick embryo. Scale bar: 50 μm. **g** Immunofluorescence for NEDD9 and phalloidin on emigrating NCCs from neural tube explants and nuclei are stained with DAPI. Polarized expression of NEDD9 is shown in pseudocolour. Line scan analysis showing the average fluorescence intensity of NEDD9 along the white dotted line ($n = 103/11$ explants). **h** Immunofluorescence for DLC1 and phalloidin on NCCs expressing control morpholino (Ctrl-MO) ($n = 65/11$ explants) or NEDD9-MO ($n = 112/12$ explants). Distribution of DLC1 expression in both treatments are shown in pseudocolour. Scale bars, 20 μm. **i** Line scan analysis showing the average fluorescence intensity of DLC1 in NEDD9-MO-treated NCCs ($n = 112/12$ explants)

It has been shown that the RHO guanine nucleotide exchange factor (GEF) Trio mediates protrusive activity of cranial NCCs through Rho GTPase activation[40] while in cell lines, GEF-H1 mediates RHOA activation in the perinuclear region[38] and leading edge[41]. Whether Trio or GEF-H1 contributes to RhoA activation in trunk NCCs remains to be determined. Nevertheless, our data reveal the asymmetric localization and activity of RhoA define the back and front polarity of NCCs for directional migration, and is spatially restricted by the polarized expression of a RhoGAP protein, DLC1.

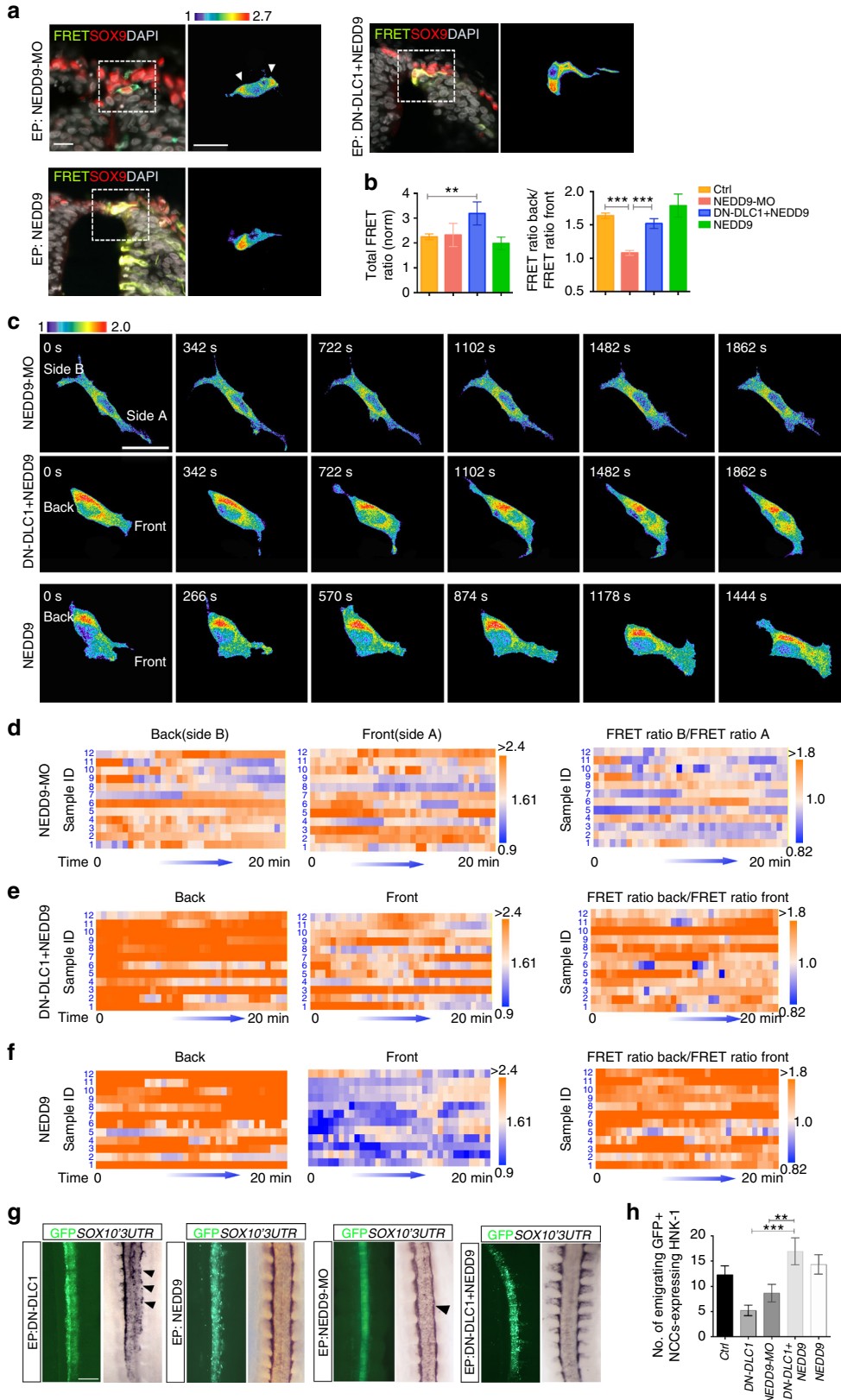

DLC1 has been well characterized as a tumor suppressor protein, suppressing tumor growth and metastasis through its RhoGAP domain to inhibit RHO signaling activity[17]. By contrast, overexpression of DLC1 resulted in inhibition of RHOA signaling and loss of apical-basal polarity in neural progenitor cells that caused their infiltration into the lumen of the neural tube. These findings coincide with an essential requirement of RHOA for the maintenance of neuroepithelial integrity[42]. In addition, the infiltration phenomenon was restricted to the dorsal neural progenitors where *RHOA* is predominantly expressed[8]. However, exogenous expression of *DLC1* gene did not change the infiltrated and non-migratory population of neural progenitors into NCC fate, indicating that DLC1 is not sufficient to induce NC specification. On the other hand, overexpression of DN-DLC1 to antagonize endogenous DLC1 activity resulted in increased RHOA activity associated with an inhibition of NC delamination, consistent with a previous report that the active form of RHO tends to maintain NCCs in the epithelial state[10]. Both in vivo and explant studies at the single cell level by FRET further demonstrated that disruption of RHOA polarity and activity (either reduced or excessive amount) by overexpression of DLC1 or DN-DLC1, respectively impeded the directionality and migratory behavior of NCCs that underlie the importance of asymmetric localization and proper level of DLC1 expression for the spatial restriction of RHOA activity to dictate NC front-back polarity. These findings are in line with the notion that the precise localization of DLC1 protein determines the spatiotemporal RHO activation pattern[17]. Moreover, our findings reveal an unexpected requirement for DLC1 in regulating NC delamination that is different from being a metastasis suppressor gene[18], implying a differential function of DLC1 between developmental and pathological context. The underlying mechanisms conferring the differences remain to be determined.

The ability of DLC1 to restrict location-specific RHO signaling depends on its subcellular location. DLC1 is recruited by specific factors to different cellular sites. In most cases, DLC1 is recruited to the membrane periphery, focal adhesions and adherens junctions that are involved in cell migration events[17]. In contrast, our data revealed that DLC1 did not show this subcellular localization but was associated with cytoplasmic NEDD9 protein expressing in a polarized manner that determines the asymmetric and cytoplasmic localization of DLC1. Previous studies showed that NEDD9 functions as an adaptor protein to assemble protein complex containing Src and FAK for inhibition of RHO/ROCK signaling in order to promote mesenchymal invasion of NC-derived melanoma cells[43]. In addition, NEDD9 has also been shown to be required for NC delamination[24] consistent with the notion that cell migratory machinery is conserved between NC and melanoma[44] but whether Src and FAK are involved in NEDD9-mediated RHO signaling restriction to regulate NC motility remains to be determined. Here, we provide new insight into NEDD9 function by demonstrating that NEDD9 is not required for the regulation of total level of RHOA activity but its interaction with DLC1 is important for the establishment of RHOA polarity between the front and back for directed migratory behavior of trunk NCCs.

SOX9 and its downstream gene *SOX10* are crucial not only for the establishment of NC identity but also for cell migration[28, 45], indicating that these two events are intimately linked. However, the transcriptional targets for SOX9 and SOX10 to regulate NC motility remain elusive. Here, we identified NEDD9 and DLC1 as the key mediators of SOX9 and SOX10 in controlling NC delamination and migration, respectively. Although SOX10 is both sufficient and necessary for *DLC1* gene expression, ectopic DLC1 expression was not detected until 12 hpt suggesting that SOX10 might activate *DLC1* expression indirectly through another factors, which remain to be identified. In addition, electroporation of SOX9 + DN-DLC1 resulted in a marked reduction of *FOXD3* expression in the early migratory NCCs to a greater extent than that of SOX10 + DN-DLC1. Considering SOX9 functions upstream of SOX10[29], these results imply that SOX9 and SOX10 might induce similar but not identical molecular events and the differences could modulate the antagonistic interaction between DN-DLC1 and ectopic *DLC1* expression, resulting in differential inhibitory effects on NC motility.

In contrast to DLC1, SOX9 but not SOX10 specifically activates *NEDD9* expression through binding to its enhancer regions. In addition, epistasis analysis reveal that NEDD9 is not required for mediating the ability of SOX9 to establish NC identity but is essential for NC delamination and migration consistent with previous studies in which downregulation of NEDD9 resulted in reduced number of migratory NCCs[24]. In conclusion, our findings provide mechanistic insight into how the transcriptional program of NC specifiers governs directional NC migratory behavior.

## Methods

**Expression vectors and morpholinos.** Full-length chick *SOX9*, *SOX10*, *DLC1*, *NEDD9* and *DN-DLC1* cDNAs were inserted upstream of an internal ribosomal entry site (IRES) and a nuclear localization sequence (nls)-tagged EGFP in a pCIG expression vector (a gift from A. McMahon, University of Southern California). A single-chain RHOA biosensor contains a fragment of Rhotekin that binds only to activated RHOA and is attached to RHOA as part of the same protein chain[46]. Full length DLC1 and DN-DLC1 were cloned into IFP2.0-C1 expression vector (a gift from C-H. Yu, the University of Hong Kong) for co-electroporation with FRET biosensor. Morpholinos (MO) for SOX9, NEDD9, SOX10, DLC1 and their corresponding MO-controls (Ctrl) were tagged with or without 3′ fluorescein and obtained from Gene Tools and their sequences are:

SOX9-MO 5′-GGGTCTAGGAGATTCATGCGAGAAA-3′,
SOX9-MO-Ctrl 5′-ATGGCCTCGGAGCTGGAGAGCCTCA-3′,
NEDD9-MO 5′-TAAGATTCTTGTACTTCATGGCTGC-3′,
NEDD9-MO-Ctrl 5′-TAACATTGTTCTACTTGATCGCTGC-3′,
SOX10-MO 5′-CCGACAGATCTTGGTCATCAGCCAT-3′
SOX10-MO-Ctrl 5′-CCCACACATCTTGCTCATGACCCAT-3′.
DLC1-MO 5′-TGGCCTCGATTTGAGTGAGGATCAT-3′
DLC1-MO-Ctrl 5′-TGCCCTCCATTTCAGTCAGCATCAT-3′
The final molar concentration of each morpholino oligonucleotide used was 0.75 mM.

**Fig. 6** Interaction between NEDD9 and DLC1 is required for RHOA polarity and directed NC migration. **a** Processed FRET signal images of cross-sections from embryos electroporated with the FRET probe plus NEDD9-MO (n = 146/17 embryos), NEDD9 (n = 168/13 embryos) or DN-DLC1 + NEDD9 (n = 152/13 embryos). NCCs are marked by SOX9 immunofluorescence and nuclei are stained with DAPI. The magnified area and selected cells are marked with dotted squares. White arrowheads indicate FRET signal. Scale bars: 20μm. **b** Quantification of the total FRET index of NCCs electroporated with the indicated constructs. Quantification of the ratio of the FRET index between the back and front of NCCs electroporated with the indicated constructs (control, n = 425/25 embryos). Mean±s.e.m.; Student's *t*-test, **$p < 0.001$; Bonferroni multiple comparison test, ***$p < 0.0001$ **c** Processed FRET signal images of time-lapse series of migrating NCCs expressing NEDD9-MO (n = 97/15 explants), DN-DLC1 + NEDD9-MO (n = 92/19 explants) and NEDD9 (n = 106/16 explants). Scale bars: 20μm. **d** Heat maps representing the FRET index as a function of time at the back or front, and the FRET ratio between the back and front of 12 selected NCCs expressing NEDD9-MO, **e** DN-DLC1 + NEDD9, and **f** NEDD9. **g** In situ hybridization of *SOX10 3'UTR* in embryos electroporated with DN-DLC1 (n = 9), NEDD9 (n = 10), NEDD9-MO (n = 10) and DN-DLC1 + NEDD9 (n = 9). Black arrowheads indicate reduced *SOX103'UTR* expression in the transfected sides. Scale bar, 10 μm. **h** Quantification of the number of GFP⁺ NCCs expressing HNK-1⁺ in embryos electroporated with the indicated constructs. Mean±s.e.m. Bonferroni multiple comparison test, **$p < 0.001$; ***$p < 0.0001$

**Chick embryos**. Fertilized chick eggs were obtained from Jinan Poultry Co. (Tin Hang Technology) and were incubated at 38 ℃ in a humidified incubator (Brinsea Octagon 250 incubator). Embryos were staged according to Hamburger and Hamilton (HH)[14]. The Committee on the Use of Live Animals in Teaching and Research, the University of Hong Kong has approved all animal experiments.

**Electroporation and neural tube explant culture**. In ovo electroporation and neural tube explants culture were performed as previously described[27, 29].

Plasmid DNA or FRET biosensor for the detection of active RHOA was injected into the lumen of HH st 11–12 neural tubes, electrodes were placed on either side of the neural tube, and electroporation was carried out using a BTX electroporator delivering five 50-ms pulses of 30 V. Electroporated embryos were allowed to

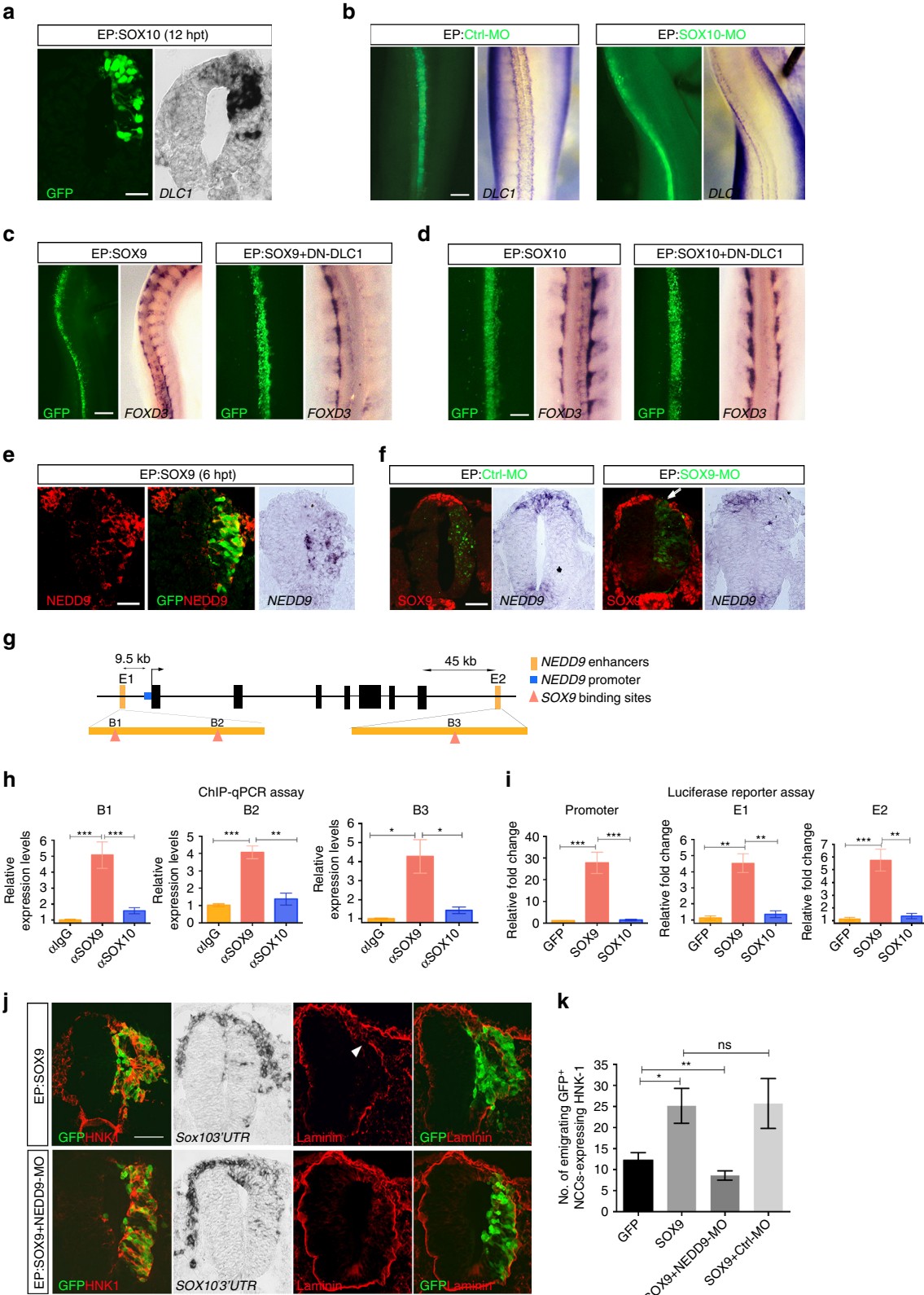

develop for 24 h post-transfection (hpt) before being processed for FRET analysis, RhoA activation assays, immunofluorescence and in situ hybridization. For neural tube explant culture, electroporated embryos were harvested at 2 hpt, and neural tubes at the level of recently formed three somites were carefully dissected following Dispase (Roche) treatment, placed onto the fibronectin-coated dishes and cultured in F12-based medium with L-glutamine (Gibco), 1% N2 supplement (Gibco), and 1% penicillin-streptomycin (Gibco) for 24 h at 37 °C and 5% CO2. Heparin-acrylic beads were washed multiple times in PBS and soaked in 50 ng/ml SDF-1 (R&D Systems) before placing adjacent to the adhered neural tubes. Subsequently, the neural tube explants were fixed for 30 min in 4% paraformaldehyde (PFA), immunostained with mouse anti-DLC1 (1:200; Santa Cruz SC-271915 or 1:500; BD Transduction Lab 612020), mouse anti-NEDD9 (1:1000; Abcam ab18056), rabbit anti-GFP (1:1000; Abcam ab6556), mouse anti-FAK (1:500; BD Transduction Lab 610088) and Alexa Fluor 568 Phalloidin (1:250; Thermo Fisher Scientific A22283) and mounted in VectaShield mounting medium with DAPI (Vector Laboratories H-1200) and photographed using a Carl Zeiss LSM 710/780 laser scanning confocal microscope in the Faculty Core Facility in Li Ka Shing Faculty of Medicine of the University of Hong Kong.

**Cell sorting and quantitative PCR**. After electroporation with *Sox10-E1:EGFP*, embryos were incubate at 37 °C until HH13 and screened for robust expression of EGFP. Embryos with weak GFP expression or at incorrect stages of development were discarded. Dissected trunks (HH13) were washed in cold PBS and dissociated with trypsin. Cells were subsequently washed, passed through a 40-μm cell strainer (BD Biosciences) and resuspended in FASC buffer at $1 \times 10^5$ cells/ml. GFP + cells were then sorted using a BD FACS Aria I Cell Sorter (BD Biosciences) with 7-AAD exclusion to eliminate dead and damaged cells. GFP positive cells were collected for quantitative PCR analysis, which was performed on an ABI PRISM Applied Bio-Systems 7000 Sequence Detection System using Power SYBY® Green Master Mix (Thermo Fisher Scientific). We designed primers specific for each Dlc1 isoform with the Primer3Plus Program: chick *DLC1 isoform 1*, forward 5′-aatgaggaagcc-gaaacgtc-3′ and reverse 5′-tcagatttcctgcgctgttg-3′; chick *DLC1 isoform 2*, forward 5′-gaaagcccctctgagaagat-3′ and reverse 5′-ccagataacatccaaggctc-3′; chick *DLC1 isoform 3* forward 5′-aagcttccttggcctcgatt-3′ and reverse 5′-tctcggaggcagcagtgag-3′; *chick GAPDH* forward 5′-attcctccacctttgatgcg-3′ and reverse 5′-tggaccatcaagtcca-caac-3′.

Real-time PCR efficiencies were determined for all sets of designed primers. Expression levels of *DLC1 isoforms* were normalized to the expression of *GAPDH*, and fold changes in mRNA expression were calculated by the $2^{-\Delta\Delta Ct}$ method. Significant changes in expression levels between isoforms were determined by unpaired Student's *t*-test.

**In situ hybridization and immunohistochemistry**. Electroporated embryos were harvested and fixed for 1 h at 4 °C in 4% paraformaldehyde in 0.1 M phosphate buffer (PB), cryoprotected with 30% sucrose in PB and sectioned at 10 μm. Immunohistochemical localization of proteins on cryosections were performed as previously described[45]. Antibodies against the following proteins were used: sheep anti-GFP (1:1000; AbD Serotec 4745-1051), guinea pig anti-SOX9 (1:1000), rabbit anti-SOX10 (1:2000, gifts from V. Lee, STEMCELL Technologies), rat anti-N-Cad (1:500; Zymed 13-2100), mouse anti-NEDD9 (1:100; Abcam), mouse anti-HNK-1 (1:100; BD Biosciences 347390), rabbit anti-SNAIL2 (1:800; Cell Signaling 9585), and rabbit anti-Laminin (1:1000; Sigma L9393). A Carl Zeiss LSM 780 Meta laser scanning confocal microscope acquired images. Whole-mount in situ hybridization using NBT/BCIP (Roche) detection was performed as described[45]. The following anti-sense ribop-robes were used: chick *SOX10* 3′UTR[27], chick *FOXD3* (I.M.A.G.E. ID 418507), chick *NEDD9* and chick *DLC1*.

**RhoA activation assay**. Following 24 hpt, 10 well-transfected neural tubes were dissected and processed for RhoA activation assays using the Biochem Kit from Cytoskeleton, Inc. Assay based on the manufacturer's instructions. In brief, equal amounts of protein from neural tube extracts were immunoprecipitated with Rhotekin-RBD followed by blotting with mouse anti-RHOA (1:1000). Horseradish peroxidase-conjugated anti-mouse immunoglobulin G (GE Healthcare) was used for the second reaction at 1:5000 dilutions. Immunocomplexes were visualized by enhanced chemiluminescence (ECL), using an ECL kit (GE Healthcare). The intensity of protein band from each sample relative to the control was quantified with ImageJ.

**Time-lapse imaging of emigrating NCCs**. Time-lapse imaging of NCCs emigrating from neural tube explants was performed on a Zeiss LSM 780 Meta laser scanning confocal microscope equipped with an incubator capable of maintaining 37 °C temperature, 95% relative humidity and 5% CO2. For cell migratory distance/speed /persistence assay, images were acquired with 10X (1.4 amplification) objective. Three to four successive images of different explants were collected every 15 min for a total period of 16 h at one time. For acquiring lifeact-mcherry and RhoA biosensor images, the 40X oil DIC lens with 1.6–1.7 amplification were applied. Activation levels of RHOA in living cells were measured by calculating the ratio of citrine-YFP FRET (emission from CFP) over CFP emission. All images for ratiometric FRET measurements were line (1) scanned at speed 4 (1024 × 1024 frame) simultaneously to avoid artefacts from cell movement between sequential CFP and FRET image acquisitions. Agron laser 458 nm were applied with maximum pinhole and laser power was set as 2 to avoid bleaching. For CFP acquisition, detector 480/40, gain (Master) 700, digital offset 0 and digital gain 2 were set up. For FRET, detector 540/30, gain (Master) 700, digital offset 0 and digital gain 1 were set up. Auto-focus was set up during time-lapse imaging.

**Image analysis**. To track distance traveled by the transfected cells over time, the Metamorph software (universal imaging) with manual tracking function was used. The distance (μm) was measured from the initial cell location near the border of the explant to the final destination. The net speed was determined by the distance traveled divided by the time of each path (h). Persistence of the cells was calculated by dividing distance, as a straight line, between the point of origin and the end point by the total distance traveled (μm). Results represent the mean ± SEM of at least 50 transfected cells from 6 explants. Significance was examined by Student's *t*-test, and a *P* value of 0.05 or less being considered significant. For polar histo-grams plot, the angle between NCC protrusions direction and single NCC migratory direction was calculated. Any actin extensions more than 4.5 μm were defined as protrusions, ADAPT plugin of ImageJ were applied for protrusion detection[47]. For FRET analysis, background subtraction, cell segmentation and ratio calculation were performed as previous described[5, 6]. Metamorph software (universal imaging) was applied for image analysis. Images were first processed with local background subtraction. The background intensity or region was defined as a set of background pixels excluding pixels with significant signal from cells or other fluorescent objects. The sum of the CFP and FRET images (after background subtraction) was then used to compute cell masks ("Threshold Image" command in Metamorph). The sum of the two channels was used for better signal to noise than either individual channel, and its value is less sensitive to differences in FRET efficiency. Cells were first crudely identified as large non-background objects with values equal to 1 and 0 elsewhere outside of the cell. Binary masks were computed locally for each cell. Each FRET ratio value was computed as the sum of the FRET intensities divided by the sum of CFP intensities over a window. No bleed through is required as CFP and Citrine-YFP are equimolar in any given pixel for this biosensor. For visual representations of ratiometeric images, pseudocolor was applied to all ratio images. A scaling factor of 1000 was specified as a multiplication

**Fig. 7** SOXE factors regulate *DLC1* and *NEDD9* expression. **a** In situ hybridization for *DLC1* on transverse sections of embryos electroporated with SOX10 at 12 hpt (*n* = 9). Scale bar, 50μm. **b** In situ hybridization for *DLC1* on embryos electroporated with Ctrl-MO (*n* = 10) or SOX10-MO (*n* = 10) at 24hpt. Black arrows indicate loss of *DLC1* expression on the transfected side of the neural tube. Scale bar, 10μm. **c** In situ hybridization for *FOXD3* on embryos electroporated with SOX9 (*n* = 9) or SOX9 + DN-DLC1 (*n* = 9) at 24hpt. Scale bar, 20 μm **d** In situ hybridization for *FOXD3* on embryos electroporated with SOX10 (*n* = 10) or SOX10 + DN-DLC1 (*n* = 10) at 24hpt. Scale bar, 10 μm. **e** Immunofluorescence for NEDD9 and in situ hybridization for *NEDD9* on transverse sections of embryos electroporated with SOX9 (*n* = 12) at 6 hpt. Scale bar, 50 μm. **f** Immunofluorescence for SOX9 and in situ hybridization for *NEDD9* on transverse sections of embryos electroporated with Ctrl-MO (*n* = 9) or SOX9-MO (*n* = 9). White arrowhead indicates loss of SOX9 expression on the transfected side. Scale bar: 50 μm. **g** Schematic diagram showing the seven exons (*black*) of chick *NEDD9* gene with promoter region in blue, −9.5 kb enhancer 1 (E1 harboring two putative *SOX9* binding motifs, B1 and B2, and + 45 kb enhancer 2 (E2) containing a putative *SOX9* binding motif, B3. **h** Fold enrichments of three independent ChIP-qPCR assays for SOX9, SOX10 and IgG (control) antibodies on B1, B2 and B3. Data represented as fold enrichments for putative *SOX9* binding motifs relative to control. Mean±s.e.m. Bonferroni multiple comparison test, \**p* < 0.05, \*\**p* < 0.001, \*\*\**p* < 0.0001. **i** Fold activation of three independent in vivo luciferase assays for reporter constructs carrying *NEDD9*-promoter, E1 or E2. Mean±s.e.m. Bonferroni multiple comparison test, \*\**p* < 0.001, \*\*\**p* < 0.0001. **j** Immunofluorescence for HNK-1, Laminin and in situ hybridization for *SOX103′UTR* on transverse sections of embryos electroporated with SOX9 or SOX9 + NEDD9-MO at 24hpt. White arrowhead indicates loss of Laminin expression on the basement membrane of the transfected neural tube. Scale bar: 50 μm. **k** Quantification of the number of emigrating GFP+ NCCs expressing HNK-1 in embryos electroporated with the indicated constructs. Mean±s.e.m. Bonferroni multiple comparison test, \**p* < 0.05; \*\**p* < 0.001; ns, not significant

**Fig. 8** Summary of the effects on RHOA activity and NC polarity following manipulation of DLC1 and NEDD9 functions. **a** SOX9 induces SOX10 to activate *DLC1* expression (dotted arrows indicate transactivation is likely to be indirect), while SOX9 directly transactivates *NEDD9* expression through binding to its enhancer regions. NEDD9 is required for the polarized expression of DLC1 and their interaction is crucial for the establishment of high RHOA activity at the back and low at the front that directs polarized NC morphology and motility. **b** Overexpression of DN-DLC1 disrupts the interaction between endogenous DLC1-NEDD9 and other cofactors through sequestration, resulting in elevation of RHOA activity throughout the cell and unpolarized NC morphology with defective motility. **c** Excessive amount of DLC1 inhibits RHOA activity throughout the cell, resulting in lack of polarized NC morphology and loss of directionality. **d** Expression of NEDD9-MO leads to lack of polarized DLC1 expression that in turn disrupts asymmetric distribution of active RHOA between the back and front of NCC which exhibits unpolarized morphology and limited motility

factor for all the ratio calculation. The ratio values were normalized to the lower scale value (1.0). To avoid low intensity pixels, bottom 5% of the total histogram distribution was excluded. Finally, ratiometeric images were subjected to a Gaussian filter of width 2 pixels (0.32 μm) to reduce pixel noise.

**Immunoprecipitation and mass spectrometry**. To identify DLC1-associated proteins for mass spectrometry, 10 chick neural tubes were micro-dissected and lysed using Pierce MS-compatible magnetic IP kit (90409). Pre-clearing of embryonic neural tube lysates was performed by incubating with protein A/G magnetic beads for 16 h at 4 °C, followed by overnight immunoprecipitation with rabbit anti-DLC1 (Abcam Ab104764) or rabbit anti-IgG as a negative control. The protein-antibody complex was then incubated with protein A/G magnetic beads for 4 h at 4 °C. The immunoprecipitated protein complex was then eluted by acidic buffer. Protein lysate was subject to Liquid chromatography-mass spectrometery analysis at the Centre for Genomic Sciences, the University of Hong Kong.

For immunoprecipitation, the DLC1-immunoprecipitated protein complex and the total input of protein lysate were immunoblotted with mouse anti-V5 (Invitrogen R96025), mouse anti-Nedd9 and mouse anti-Dlc1.

All uncropped western blots can be found in Supplementary Fig. 10.

**Chromatin immunoprecipitation**. ChIP was performed based on the protocol[19]. Dorsal neural tubes were dissected in Ringer's solution from 20 st 11–12 chick embryos and transferred to 1 ml isotonic buffer [0.5% Triton X-100, 10 mM Tris-HCl, pH 7.5, 3 mM CaCl$_2$, 0.25 M sucrose, protease inhibitor table (Complete Protease Inhibitor EDTA-free, Roche), 1 mM DTT, and 0.2 mM PMSF] on ice. Tissue was homogenized and cells cross-linked by adding formaldehyde to a final concentration of 1% and nutated for 10 min at room temperature. Glycine (final concentration of 125 mM) was added to stop the cross-linking reaction. The cross-linked cells were washed three times and cell pellets were re-suspended in isotonic buffer and nuclei isolated using homogenizer, washed, and lysed in SDS lysis buffer (1% SDS, 10 mM EDTA, 50 mM Tris-HCl, pH 8.0) for 10 min. The lysate was then diluted 3-fold with ChIP dilution buffer (0.01% SDS, 1.2 mM EDTA, 16.7 mM Tris-HCl pH 8.0, 167 mM NaCl, 1 mM DTT, 0.4 mM PMSF, and protease inhibitors) and one half of chromatin was sheared into ~ 200 to 500 bp DNA fragments using Diagenode Bioruptor® Plus sonicator. The sonicated chromatin was immunoprecipitated with Dynal bead protein A (Invitrogen) pre-absorbed with anti-SOX9, anti-SOX10 or control rabbit anti-IgG (Abcam). Samples were washed, eluted and reverse cross-linked. The precipitated DNA fragments were purified and quantified by qPCR with primers specific for NEDD9-p, E1 and E2 regions (Supplementary Fig. 6d). Each sample was run in triplicate and the results were quantified using the ΔΔC$_t$ method. Analysis was done using Applied Biosystem's instructions.

**In vivo luciferase assay**. The luciferase reporter assay was performed based on the manufacturer's protocol from Dual-Luciferase® Reporter Assay System (Promega). Briefly, *NEDD9-p*, *E1*- or *E2* enhancers driven luciferase reporters were mixed with Renilla and SOX9 or SOX10 expression construct and electroporated into the chick neural tube. The trunk parts of eight well-transfected chick embryos from each treatment were harvested 24 hpt, lysed and assayed for firefly luciferase activity (Microplate Luminometer LB96V, EG&G BERTHOLD). Experiments were performed in triplicate. Statistical analysis was performed using Student's *t*-test.

**Data availability**. The authors declare that all data supporting the findings of this study are available within the article and its supplementary information files or from the corresponding author upon reasonable request. The ChIP-qPCR data have been deposited in Figshare (10.6084/m9.figshare.5324431)[48].

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

## Acknowledgements

We thank J. Briscoe for helpful discussion and comments on the manuscript, J. Guo, C. Lai and C.-H. Yu for their assistance and advice on FRET imaging and analysis, and M. Way for providing LifeAct-mCherry construct. Irene Ng is Loke Yew Professor in Pathology. This work was supported by grants from the Research Grants Council and University Grants Council of Hong Kong (ECS_27100314), (X_HKU708/14), (GRF_17110715) to M.C. and J.AI.L. and (AoE/M-04/04), (T12-708/12-N) to K.S.E.C.

## Author contributions

J.AI.L. and M.C. conceived and designed the experiments and wrote the manuscript. J.AI.L. and M.C. performed and analyzed most of the experiments, except in situ hybridization, immunofluorescence, cell sorting and quantitative PCR, which were carried out by Y.X.R. and M.P.L.C. ChIP and reporter assays were carried out by M.N.H. and M-H.W. In silico analysis of *NEDD9* regulatory regions was carried out by B.N. DLC1 constructs were provided by L.K.C. and I.O.L.N. K.S.E.C. designed ChIP and reporter assays. R.S. performed shotgun proteomics analysis. L.H. generated the FRET construct and analyzed data.

## Additional information

**Competing interests:** The authors declare no competing financial interests.

