## [Peer Review File · Nature Communications]

Reviewers' Comments:

Reviewer #1 (Remarks to the Author)

The manuscript uses a FRET reporter for RhoA to probe RhoA activity in explanted neural crest cells. The authors make the argument that selective spatial inhibition of RhoA by Dlc1 allows activity to accumulate in otherwise symmetrical cells, leading to a symmetry break which drives migration.

Major points

One general concern is that FRET ratios are displayed in different LUTs in virtually every panel of every figure. This makes comparison of RhoA activity levels between images impossible. For example, in figure 6 a Nedd9-MO cells are compared to Dn-Dlc1+Nedd9 cells. A reader might reasonably assume based on the false-color display that red regions in the top and bottom cells represent similar levels of activity, but they do not. A reader might want to compare activity levels between cells in different figures, but they can not do this based on the way the images are displayed. While the LUT for each panel carries its own numerical scale, its not reasonable to figure out in your head how the maximum activity of the Nedd9-MO cell compares to the average activity of the Dn-Dlc1+Nedd9 cell. The effect of stretching every image to use the full purple-to-red colour scale is to conceal an aspect of cellular heterogeneity. This homogenised representation of RhoA activity between cells fails to capture, indeed overly simplifies, biological complexity.

Fig 1a-f: these figures show that RhoA activity is polarised in migrating cells. This suggests that non-migrating cells do not display polarised RhoA activity. But is this the case; is RhoA activity not polarised in non-migrating cells? Non-migrating cells obviously don't have a front/back axis defined by their direction of migration, but they might still be elongated, which would provide an axis along which to compare RhoA activity.

Fig 1j & SV3: video shows polarisation of RhoA but does not clearly show resulting protrusion and direction of migration. Initial accumulation of Rho activity in the upper left of the cell prior to retraction of the long tail suggests cell should migrate down and to the right, but location of Rho activity again changes. So the most one can say is that sometimes localised RhoA activity indicates the subsequent direction of migration. SV4 is not convincing on this point either: RhoA activity is generally high in the perinuclear area and the directions of the white arrows indicating activity and direction are all over the place. P6L2: change "highly predictive of" to "sometimes indicates"

Fig 2 g&h: Please quantify high and low signal intensity. Assuming this is a linear LUT, please express the High signal level as a multiple of the Low signal level. See also Fig 5 f and g.

Fig 2 k-m: Its fine to say that RhoA activity is high in the back whereas Dlc1 localises to the front. But this figure goes a bit far by comparing localisation patterns in different groups of cells (RhoA transfected or DLC stained, unless i've got it wrong). At the very least panel m should be removed.

Fig 3 d&e : The effects of DLC1 and DN-Dlc1 over-expression on RhoA activity (d) are generally convincing in that they lead to loss of cell polarity, but the analysis in panel e muddies the water. First, according to the author's hypothesis DN-Dlc1 cells should not migrate unless they are able to localise RhoA activity to one region of the cell. But cells 10 and 12 appear to contradict this for the 20 minute observation period, and other cells appear to contradict this at least temporarily. I would remove panel e entirely because the main point here is overall loss of polarity and normal migration.

Fig 6: The text states (P13L) "Surprisingly total RhoA activity in cells treated with Nedd9-MO was comparable to the control suggesting that loss of Nedd9 function has no direct impact on RhoA activity." -Which data have been used to draw this conclusion? Was total RhoA activity assessed from FRET images? How was this calculated? I don't think it is logical to assume that removing a signal which localises a Rho inhibitor should have no impact on the size of the pool of active RhoA (however that is being assessed).

Minor points:

Where is information on the spatial and temporal calibration of the Supplementary Videos presented?

Reviewer #2 (Remarks to the Author)

The manuscript by Liu et al. addresses the spatial regulation of the RhoA GTPase during directional delamination and migration of neural crest cells in chick neural tube explants. The authors claim that the Rho gradient established in these polarized cells with high levels at the rear is dependent upon the GTPase-activating protein DLC1, for which they show localization to the leading edge of the cells, whereby Rho-GTP levels are kept low at this site. The authors further claim that the adaptor protein Nedd9, which was reported to be involved in neural crest cell migration, recruits DLC1 to the cell front and thereby is responsible for polarized RhoA activity. Finally, the functional contribution of the Nedd9-DLC1 complex to neural crest delamination and migration is supported by their joint transcriptional upregulation downstream of SOX9/SOX10.

As such, this is a novel and very interesting study, as the spatiotemporal regulation of cellular RhoA activity is still poorly understood. In addition, whereas the function of DLC1 as a tumor suppressor has been studied quite extensively, there is still little known about its normal physiological function in developmental processes and transcriptional regulation. However, although the authors use sophisticated techniques and in general the data are convincing, there are some essential experimental controls missing that are required to support the conclusions drawn by the authors. In addition, the manuscript should be proof-read by a native English speaker prior to submission of a revised version.

Major comments

My major concern is that the role of DLC1 in neural crest cell polarity and migration is deduced only from overexpression experiments of either full-length DLC1 or a presumably dominant-negative variant of DLC1 (DN-DLC1), which consists of the DLC1 N-terminus and lacks the C-terminal GAP and START domains. The authors show that ectopic expression of DN-DLC1 increases RhoA activity, but they cannot be sure that this really is due to the competition with endogenous DLC1. For example, the family members DLC2 and DLC3 possess partially conserved N-terminal regions and might also be displaced by DN-DLC1, as would many other endogenous proteins that interact with a common set of adaptors. It is absolutely necessary to demonstrate that DLC1 is required for neural crest cell polarity and migration not only by expressing DLC1 or DN-DLC1 but also by suppressing endogenous DLC1 expression using morpholinos.

The authors claim that DN-DLC1 acts in a dominant-negative manner. However, they do not provide any mechanistic explanation for this activity, especially considering that this construct was originally shown to compete with focal adhesion localization. This is believed to involve binding of tensin adaptor proteins. Does DN-DLC1 interact with Nedd9? Does DN-DLC1 interfere with the endogenous Nedd9-DLC1 interaction? Which domains in DLC1 and Nedd9 are required for the interaction? Is the interaction direct? The answers to these questions are important to interpret the data shown in Figure 6.

In Figure 6, the controls showing the effect of Nedd9 overexpression alone are missing. Perhaps ectopic expression of Nedd9 alone is sufficient for the rescue, simply by additive effects? The

figure legend and labeling of this figure is inconsistent: Is DN-DLC1 coexpressed with Nedd9 or Nedd-MO?

Minor comments

How was the specificity of the DLC1 antibody staining controlled (see Figure 2g)?

Reviewer #3 (Remarks to the Author)

In this paper, the authors examine the molecular signals that underlie how trunk neural crest cells establish and regulate their polarity during emigration from the neural tube and early migration events. The authors first characterize dynamic RhoA activity in delaminated and migrating neural crest cells in vitro using a fluorescence biosensor and time-lapse imaging, then identify the RhoGAP Dlc1 as a potential regulator of its activity. Through an experimental series of *gof/lof* of Dlc1 and identification of Nedd9 as a Dlc1-interacting protein, the authors examine the function of Dlc1 and propose a potential regulatory circuit that includes Sox9/10 driving Nedd9/Dlc1 to regulate neural crest cell polarity.

Overall, many of the molecular signals in this study have already been identified and investigated in either the neural crest or cancer metastasis models, including in vivo results on the function of RhoA in neural crest cell migration and functional studies of Dlc1 and Nedd9 in cancer metastasis. The novelty in this paper is their observations of dynamic RhoA activity, albeit in vitro, and identification of Dlc1/Nedd9 signaling specifically in the neural crest model. The authors attempt to bring together several pieces of information both from in vitro and some in vivo that together provide a plausible mechanistic explanation for neural crest cell polarity during delamination and early emigration. Unfortunately, there are several substantive concerns with both the approach and data that significantly limit the results of this study and potential impact of to the neural crest and broader fields of directed cell migration/cancer cell invasion. Thus, the study needs significant revision and is also more suited to a specialty journal.

First, the in vitro assay from which the authors measure RhoA activity and cell behavior dynamics in *gof/lof* of Dlc1 etc is not a good system to dissect cell polarity and direction migration since there is no presence of a directional signal. Previous results have studied RhoA function in vivo during neural crest cell migration in two or more distinct embryo model systems (Matthews et al., 2008; Carmona-Fontaine et al., 2008; Theveneau and Mayor, 2010; Rupp and Kulesa, 2007). In in vitro neural tube explants, cells simply spread out into unoccupied areas and are not a true representation of directed migration. A much stronger case would be to introduce a neural crest cell chemoattractant in vitro such as Sdf1, Nrg1 that have been shown in vivo to direct neural crest cell migration (Olesnick-Killian et al., 2009; Saito et al., 2012; Theveneau et al, 2013). In the absence of in vivo imaging or a directional signal in vitro, the authors suggest that RhoA activity is asymmetric during neural crest cell delamination and migration and rely on visualization of events in vitro of neural tube explants. However, in their static and time-lapse images, it appears that RhoA activity is 'not' highly predictive of the future back-front polarity and directional neural crest cell migration. Rather, their data show that RhoA activity appears in the back of the cell 'after' the cell has established a choice of and moved in a particular direction (shown by the thin cytoplasmic tail dragged behind the cell and localization of RhoA on the lateral side but not back of cell). Furthermore, the authors analyze an extremely low number of cells ($n=12/m=1$ explant) in the RhoA activity and this is indicative throughout the paper. To be realistic in most cell behavior analyses and for statistical measurements, they should analyze at least 500 cells in multiple (at least 5) explanted neural tubes.

Second, the authors present expression data of Dlc1 and claim this is localized to leading edge protrusions during delamination and migration. They present clear in vitro data (neural tube explant culture), but their in vivo data is both vague and under-represented (Fig. Sup 2). Why are there so few labeled neural crest cells and why is there no overlap in the Dlc1/GFP signal? Surely

the GFP signal should be throughout the cell and not localized to the rear majority of the cell. Also, Fig Sup 2 shows Dlc1 is not present in all cell protrusions? Why did the authors not push the expression analysis in vivo?

Third, manipulation of Dlc1 and subsequent affects on neural crest cell behaviors are measured in vitro. A stronger case would be to make these measurements in vitro in the presence of a guidance signal or in vivo, otherwise the overall speculation that Dlc1 regulates cell polarity and requirement for directional delamination is not conclusive. Furthermore, cells in which Dlc1 is over-expressed appear to be more amoeboid in shape but are not shown to migrate. This makes it difficult to evaluate RhoA signaling since the cells are not moving in any direction nor have significant protrusive activity. It is also difficult to argue that neural crest cells require precise spatial activity of Dlc1 for restricting RhoA activity low at the front since ctrl cells show high localized RhoA activity in active cell protrusions.

Fourth, it is difficult to judge whether Nedd9 interacts with Dlc1 since Nedd9 MO cells look more polarized than Ctrl-MO cells (Fig. 5g) and there is no quantitation of fluorescence as in all previous figures.

Below please find our point-by-point response to reviewers' comments with changes highlighted in **red** in the revised manuscript (the reviewer's comments are in *italics*).

Reviewer #1 (Remarks to the Author):

The manuscript uses a FRET reporter for RhoA to probe RhoA activity in explanted neural crest cells. The authors make the argument that selective spatial inhibition of RhoA by Dlc1 allows activity to accumulate in otherwise symmetrical cells, leading to a symmetry break which drives migration.

Major points

1. One general concern is that FRET ratios are displayed in different LUTs in virtually every panel of every figure. This makes comparison of RhoA activity levels between images impossible. For example, in figure 6 a Nedd9-MO cells are compared to Dn-Dlc1+Nedd9 cells. A reader might reasonably assume based on the false-color display that red regions in the top and bottom cells represent similar levels of activity, but they do not. A reader might want to compare activity levels between cells in different figures, but they can not do this based on the way the images are displayed. While the LUT for each panel carries its own numerical scale, its not reasonable to figure out in your head how the maximum activity of the Nedd9-MO cell compares to the average activity of the Dn-Dlc1+Nedd9 cell. The effect of stretching every image to use the full purple-to-red colour scale is to conceal an aspect of cellular heterogeneity. This homogenised representation of RhoA activity between cells fails to capture, indeed overly simplifies, biological complexity.

Author Response:

We have normalized the FRET ratio LUT scale to the lower scale value (1.0) to allow easier comparisons among our FRET ratio images presented in figures 1-6. However, the LUT scale for DN-Dlc1 in figure 3c and 4h is out of the range and does not lie within the scale for control and Dlc1 that makes it impossible to have a common LUT scale for comparison between three treatments. We have described a method for the calculation of FRET ratiometric images on page 4 in the Methods.

2. Fig 1a-f: these figures show that RhoA activity is polarised in migrating cells. This suggests that non-migrating cells do not display polarised RhoA activity. But is this the case; is RhoA activity not polarised in non-migrating cells? Non-migrating cells obviously don't have a front/back axis defined by their direction of migration, but they might still be elongated, which would provide an axis along which to compare RhoA activity.

Author Response:

In Fig S1a,b, we examined RhoA activity in non-migrating neural tube cells, which do not have distinct front-back polarity axis. Consistently, neuroepithelial cell with an elongated morphology exhibits moderate level of RhoA activity, which is uniformly distributed throughout the cell (Fig. S1b). We have described the results on page 5.

3. Fig 1j & SV3: video shows polarisation of RhoA but does not clearly show resulting protrusion and direction of migration. Initial accumulation of Rho activity in the upper left of the cell prior to retraction of the long tail suggests cell should migrate down and to the right, but location of Rho activity again changes. So the most one can say is that sometimes localised RhoA activity indicates the subsequent

direction of migration. SV4 is not convincing on this point either: RhoA activity is generally high in the perinuclear area and the directions of the white arrows indicating activity and direction are all over the place. P6L2: change “highly predictive of” to “sometimes indicates”

Author Response:

We have replaced Fig. 1j & SV3 with a new Fig. 1i and Supplementary video 2, respectively. The results clearly show initial polarization of RhoA activity along the front-back axis and the subsequent re-localization of RhoA activity to the prospective cell rear is synchronized with the establishment of new membrane protrusions at the cell front as neural crest cells underwent directional change in response to SDF-1 (Fig. 1i,j and Supplementary Video 2). We have also replaced SV4 with a new video (now Supplementary Video 3) which clearly shows that NCC with a front-back polarized morphology gradually acquired elevated RhoA activity at the front (A) which eventually became the back of the cell following cell repolarization together with the formation of a new membrane protrusion at the front pointing toward SDF-1 (B) (Fig. 1l and m and Supplementary Video 3). We have changed our conclusion into “asymmetry of RhoA activity **indicates** the future back-front polarity and directional migration of NCCs” on page 7. We have described the results for new figures on pages 6 and 7.

4. Fig 2 g&h: Please quantify high and low signal intensity. Assuming this is a linear LUT, please express the High signal level as a multiple of the Low signal level. See also Fig 5 f and g.

Author Response:

We have quantified signal intensity in fig 2g,h and fig 5g,h and expressed the High signal level as a multiple of the Low signal level ranging from 395 to 4000.

5. Fig 2 k-m: Its fine to say that RhoA activity is high in the back whereas Dlc1 localises to the front. But this figure goes a bit far by comparing localisation patterns in different groups of cells (RhoA transfected or DLC stained, unless i’ve got it wrong). At the very least panel m should be removed.

Author Response:

The reviewer is correct. We compared co-localization patterns between RhoA activity and Dlc1 fluorescence intensity from the leading cell edge to perinuclear region of cytoplasm in different groups of cells (Fig. 2k), and confirmed a negative correlation with high Dlc1 expression in the region of low RhoA activity and vice versa (Fig. 2l,m) as described in page 9. As suggested by the reviewer, we have removed panel m to avoid confusion.

6. Fig 3 d&e: The effects of DLC1 and DN-Dlc1 over-expression on RhoA activity (d) are generally convincing in that they lead to loss of cell polarity, but the analysis in panel e muddies the water. First, according to the author’s hypothesis DN-Dlc1 cells should not migrate unless they are able to localise RhoA activity to one region of the cell. But cells 10 and 12 appear to contradict this for the 20 minute observation period, and other cells appear to contradict this at least temporarily. I would remove panel e entirely because the main point here is overall loss of polarity and normal migration.

Author Response:

As suggested by the reviewer, we have removed panel e to avoid confusion.

7. Fig 6: The text states (P13L) “Surprisingly total RhoA activity in cells treated with Nedd9-MO was comparable to the control suggesting that loss of Nedd9 function has no direct impact on RhoA activity.” -Which data have been used to draw this conclusion? Was total RhoA activity assessed from FRET images? How was this calculated? I don’t think it is logical to assume that removing a signal which localises a Rho inhibitor should have no impact on the size of the pool of active RhoA (however that is being assessed).

Author Response:

We have included western blot data below showing that Nedd9-MO treated cells did not affect the level of Dlc1 protein expression but disrupted the polarized expression of Dlc1 with random distribution throughout the cell as shown by line scan analysis (Fig. 5h,i). This mislocalization of Dlc1 protein caused aberrant distribution of RhoA activity around the nucleus of Nedd9-MO-treated cells (Fig. 6a,c and Supplementary Video 10), resulting in lowering the ratio of the FRET index between back and front without affecting the size of the pool of active RhoA (total FRET) as shown in Fig. 6b-d. We have described the results on page 16 and 17.

*Minor points:*

Where is information on the spatial and temporal calibration of the Supplementary Videos presented?

Author Response:

We have included the LUT scale in all the Supplementary Videos presented.

Reviewer #2 (Remarks to the Author):

The manuscript by Liu et al. addresses the spatial regulation of the RhoA GTPase during directional delamination and migration of neural crest cells in chick neural tube explants. The authors claim that the Rho gradient established in these polarized cells with high levels at the rear is dependent upon the GTPase-activating protein DLC1, for which they show localization to the leading edge of the cells, whereby Rho-GTP levels are kept low at this site. The authors further claim that the adaptor protein Nedd9, which was reported to be involved in neural crest cell migration, recruits DLC1 to the cell front and thereby is responsible for polarized RhoA activity. Finally, the functional contribution of the Nedd9-DLC1 complex to neural crest delamination and migration is supported by their joint transcriptional upregulation downstream of SOX9/SOX10.

As such, this is a novel and very interesting study, as the spatiotemporal regulation of cellular RhoA activity is still poorly understood. In addition, whereas the function of DLC1 as a tumor suppressor has been studied quite extensively, there is still little known about its normal physiological function in developmental processes and transcriptional regulation. However, although the authors use sophisticated techniques and in general the data are convincing, there are some essential experimental controls missing that are required to support the conclusions drawn by the authors. In addition, the manuscript should be proof-read by a native English speaker prior to submission of a revised version.

Major comments

1. *My major concern is that the role of DLC1 in neural crest cell polarity and migration is deduced only from overexpression experiments of either full-length DLC1 or a presumably dominant-negative variant of DLC1 (DN-DLC1), which consists of the DLC1 N-terminus and lacks the C-terminal GAP and START domains. The authors show that ectopic expression of DN-DLC1 increases RhoA activity, but they cannot be sure that this really is due to the competition with endogenous DLC1. For example, the family members DLC2 and DLC3 possess partially conserved N-terminal regions and might also be displaced by DN-DLC1, as would many other endogenous proteins that interact with a common set of adaptors. It is absolutely necessary to demonstrate that DLC1 is required for neural crest cell polarity and migration not only by expressing DLC1 or DN-DLC1 but also by suppressing endogenous DLC1 expression using morpholinos.*

Author Response:

We electroporated morpholino targeting Dlc1 isoform 3 (Dlc1-MO) into the caudal hemineural tube of st 10-11 and showed dosage-dependent reduction in endogenous Dlc1 protein expression at 24hpt as shown in Fig S3a. We also observed a similar reduction of Dlc1 expression in Dlc1-MO-treated cultured neural crest cells, which appeared round in shape without front-back polarity like DN-Dlc1 expressing cells (Fig. S3b, Fig. 3c). In agreement with this, cells expressing Dlc1-MO resulted in reduced expression of *FoxD3*⁺ migrating NCCs compared to the untransfected side and the Ctrl-MO-treated embryos (Fig. S3c). These results demonstrate functional requirements for Dlc1 in the establishment of NC polarity and delamination. We have described the results on page 11-12.

2. The authors claim that DN-DLC1 acts in a dominant-negative manner. However, they do not provide any mechanistic explanation for this activity, especially considering that this construct was originally shown to compete with focal adhesion localization. This is believed to involve binding of tensin adaptor proteins. Does DN-DLC1 interact with Nedd9? Does DN-DLC1 interfere with the endogenous Nedd9-DLC1 interaction? Which domains in DLC1 and Nedd9 are required for the interaction? Is the interaction direct? The answers to these questions are important to interpret the data shown in Figure 6.

Author Response:

We performed co-immunoprecipitation assay in chick embryonic neural tube lysates containing V5-DN-Dlc1 or C-terminally truncated mutants of V5-DN-Dlc1 followed by western blotting with anti-Nedd9 to examine whether DN-Dlc1 could interact with endogenous Nedd9 and the domains involved for the interaction. In fig 5e, the results revealed the ability of DN-Dlc1 to interact with Nedd9 via its focal adhesion-targeting (FAT) region but not the SAM domain. Based on these results together with the fact that endogenous Nedd9 is in limiting amount to associate with Dlc1, it is reasonable to assume that overexpression of DN-Dlc1 is able to interfere with the endogenous Nedd9-Dlc1 interaction through Nedd9 sequestration. Indeed, overexpression of DN-Dlc1 or DN-Dlc1 (1-320), but not the SAM domain, was able to reduce neural crest delamination (Fig S4c). We have described the results on page 15. Altogether, these results provide an explanation as to why overexpression of Nedd9 is able to restore differential RhoA activity, polarized morphology and directional migration in DN-Dlc1 expressing cells as shown in Figure 6.

3. In Figure 6, the controls showing the effect of Nedd9 overexpression alone are missing. Perhaps ectopic expression of Nedd9 alone is sufficient for the rescue, simply by additive effects? The figure legend and labeling of this figure is inconsistent: Is DN-DLC1 coexpressed with Nedd9 or Nedd-MO?

Author Response:

In fig 6, we have included the control of Nedd9 overexpression alone that showed normal onset of neural crest delamination without alteration of polarized RhoA activity and directional migratory behavior (Supplementary Video 12).

We apologize for mislabeling of the figure legend, which has been amended to DN-Dlc1+Nedd9 instead of Nedd9-MO.

Minor comments

How was the specificity of the DLC1 antibody staining controlled (see Figure 2g)?

Author Response:

In Fig S3b, the level of Dlc1 expression in Dlc1-MO-treated cultured neural crest cells was significantly reduced compared to untransfected cells in which Dlc1 expression remains, indicating the specificity of the DLC1 antibody.

The manuscript has been proofread by a native English speaker.

Reviewer #3 (Remarks to the Author):

*In this paper, the authors examine the molecular signals that underlie how trunk neural crest cells establish and regulate their polarity during emigration from the neural tube and early migration events. The authors first characterize dynamic RhoA activity in delaminated and migrating neural crest cells in vitro using a fluorescence biosensor and time-lapse imaging, then identify the RhoGAP Dlc1 as a potential regulator of its activity. Through an experimental series of *gof/lof* of Dlc1 and identification of Nedd9 as a Dlc1-interacting protein, the authors examine the function of Dlc1 and propose a potential regulatory circuit that includes Sox9/10 driving Nedd9/Dlc1 to regulate neural crest cell polarity.*

Overall, many of the molecular signals in this study have already been identified and investigated in either the neural crest or cancer metastasis models, including in vivo results on the function of RhoA in neural crest cell migration and functional studies of Dlc1 and Nedd9 in cancer metastasis. The novelty in this paper is their observations of dynamic RhoA activity, albeit in vitro, and identification of Dlc1/Nedd9 signaling specifically in the neural crest model. The authors attempt to bring together several pieces of information both from in vitro and some in vivo that together provide a plausible mechanistic explanation for neural crest cell polarity during delamination and early emigration. Unfortunately, there are several substantive concerns with both the approach and data that significantly limit the results of this study and potential impact of to the neural crest and broader fields of directed cell migration/cancer cell invasion. Thus, the study needs significant revision and is also more suited to a specialty journal.

*1. First, the in vitro assay from which the authors measure RhoA activity and cell behavior dynamics in *gof/lof* of Dlc1 etc is not a good system to dissect cell polarity and direction migration since there is no presence of a directional signal. Previous results have studied RhoA function in vivo during neural crest cell migration in two or more distinct embryo model systems (Matthews et al., 2008; Carmona-Fontaine et al., 2008; Theveneau and Mayor, 2010; Rupp and Kulesa, 2007). In in vitro neural tube explants, cells simply spread out into unoccupied areas and are not a true representation of directed migration. A much stronger case would be to introduce a neural crest cell chemoattractant in vitro such as Sdf1, Nrg1 that have been shown in vivo to direct neural crest cell migration (Olesnicky-Killian et al., 2009; Saito et al., 2012; Theveneau et al., 2013). In the absence of in vivo imaging or a directional signal in vitro, the authors suggest that RhoA activity is asymmetric during neural crest cell delamination and migration and rely on visualization of events in vitro of neural tube explants. However, in their static and time-lapse images, it appears that RhoA activity is 'not' highly predictive of the future back-front polarity and directional neural crest cell migration. Rather, their data show that RhoA activity appears in the back of the cell 'after' the cell has established a choice of and moved in a particular direction (shown by the thin cytoplasmic tail dragged behind the cell and localization of RhoA on the lateral side but not back of cell). Furthermore, the authors analyze an extremely low number of cells ($n=12/m=1$ explant) in the RhoA activity and this is indicative throughout the paper. To be realistic in most cell behavior analyses and for statistical measurements, they should analyze at least 500 cells in multiple (at least 5) explanted neural tubes.*

Author Response:

Thanks for the reviewer's constructive comments, which we agree to investigate RhoA activity both in vivo and in vitro neural tube explants in the presence of chemoattractant, SDF-1. We performed FRET analysis *in vivo* by electroporating a RhoA single chain FRET-based biosensor into the caudal neural tube of chick embryos at stage 11-12. At 24 hour post-transfection (hpt), electroporated embryos at stage 15-16 were harvested for FRET analysis on cross section of thoracic neural tube where neural crest delamination and migration are ongoing. FRET imaging and measurement of the FRET index between the back and front of NCCs revealed high RhoA activity in the cytoplasm of the cell rear relative to low or fluctuating RhoA activity in membrane protrusions at the leading front of delaminating/early- and late-migrating neural crest cells (Fig. 1a-d). In contrast, we detected moderate level and uniform distribution of RhoA activity throughout neuroepithelial cells in the neural tube (Fig. S1a,b). These results indicate that NCCs acquire differential RhoA activity in subcellular localization as they undergo directional delamination and migration. Consistent with *in vivo* observations, time-lapse imaging of neural tube

explants electroporated with FRET biosensor showed that RhoA activity was highly enriched at the cell rear and also dynamically localized in membrane protrusions at the leading edge of polarized neural crest cells undergoing directional delamination from the explants (Fig. 1e and Supplementary Video 1). We have described the results on pages 5 and 6.

To further interrogate whether this differential RhoA activity between the back and front is maintained as neural crest cells change in migratory direction, emigrating neural crest cells from neural tube explants were exposed to beads coated with stromal cell derived factor (SDF-1), which is a chemoattractant for trunk neural crest cells, to mimic the *in vivo* environment as suggested by the reviewer. After addition of SDF-1 coated beads on the other side of emigrating NCCs, we observed initial polarized RhoA activity along the front-back axis and the subsequent re-localization of high RhoA to the prospective cell rear is synchronized with the establishment of new membrane protrusions at the cell front as NCCs underwent directional change in response to SDF-1 (Fig. 1i and j and Supplementary Video 2). Quantification of the FRET index between the back and front over time revealed the maintenance of high RhoA activity at the cell rear even when NCCs changed their direction of movement. Thus, pre-existing asymmetrical localization of RhoA activity marks the cell's eventual direction of polarization. To further correlate asymmetric RhoA activity with cell polarization, we examined RhoA dynamics in a population of emigrating NCCs, which undergo front-back switch in response to SDF-1 on the opposite side of the cell. Time-lapse FRET imaging showed that NCC with a front-back polarized morphology exhibited elevated RhoA activity at the front which eventually became the back of the cell following cell repolarization associated with the formation of a new membrane protrusion at the front pointing towards SDF-1 (Fig. 1l and m and Supplementary Video 3). Quantification of the FRET activity between back and front over time revealed that redistribution of differential RhoA activity preceded front and back polarity switch (Fig. 1n). Altogether, *in vitro* neural tube explant studies further consolidate *in vivo* observations that existing asymmetry of RhoA activity indicates the future back-front polarity and directional migration of trunk NCCs. We have described the results on pages 6 and 7.

We also studied the effects of overexpression of Dlc1 and DN-Dlc1 on RhoA activity; neural crest polarity and migration in both chick embryos (Fig. 3) and neural crest explant culture in the presence of SDF-1 (Fig. 4g-h). First, we overexpressed FRET biosensor together with Dlc1 into the caudal hemineural tube of st 10-11 chick embryos and analyzed at 24 hpt for FRET signals on sections. The results showed that RhoA activity was significantly reduced or barely detectable in Dlc1 overexpressing NCCs, which exhibited lack of front-back polarity axis (Fig. 3c). In addition examination of NC markers revealed that overexpression of Dlc1 did not induce ectopic expression of Sox9, Sox10 and HNK-1 in the neuroepithelium NC formation, suggesting that Dlc1 is not sufficient to trigger NC formation. However, expression of these markers in migratory NCCs was reduced on the transfected side (Fig. 3e-f' and o). Strikingly, neuroepithelial cells overexpressing Dlc1 were observed delaminating not only from the basal surface of the dorsal neural tube where laminin was lost but also into the lumen (Fig. 3g,g'). Consistently, N-Cadherin (N-Cad) expression was lost at the apical surface of dorsal neuroepithelium (Fig. 3h,h'), indicating that overexpression of Dlc1 disrupted apical-basal polarity of dorsal neural tube cells. These data suggest that high level of Dlc1 expression reduced RhoA activity, resulting in aberrant polarity of premigratory NCCs that delaminate into the neural tube lumen instead of following their normal migratory route. We have described the results on pages 9 and 10.

We then examined whether dominative negative inhibition of Dlc1 affected RhoA activity, neural crest polarity and migratory behavior. By contrast to Dlc1, overexpression of DN-Dlc1 *in vivo* resulted in a marked elevation of RhoA activity, which was evenly distributed throughout the cytoplasm of delaminating NCCs without distinct subcellular localization and cells appeared round in shape (Fig. 3c). Quantitative analysis of total FRET activity *in vivo* revealed that, compared to vector control, the overall levels of RhoA activation were significantly increased ($p < 0.001$) and reduced ($p < 0.01$) in NCCs overexpressing DN-Dlc1 and Dlc1, respectively (Fig. 3d). Analysis of NC markers revealed that overexpression of DN-Dlc1 resulted in markedly fewer migratory NCCs expressing HNK-1 compared to the contralateral side (Fig. 3i-j' and o). Consistently, the amount of early migratory NCCs expressing Sox9 and Sox10 were reduced on the transfected side but their expression in the premigratory domain

remains unaltered, indicating that NC identity was not affected by overexpression of DN-Dlc1 (Fig. 3k,l). In contrast to Dlc1, the majority of NCCs expressing DN-Dlc1 remained in the neuroepithelium without disrupting apical-basal polarity (Fig. 3m,n). To further demonstrate that Dlc1 is required for NC polarity and delamination, we also electroporated a fluorescein-tagged morpholino targeting Dlc1 isoform 3 (Dlc1-MO) into the caudal hemineural tube of st 10-11 embryos. At 24 hpt, Dlc1 protein was significantly diminished using higher dosage of Dlc1-MO, whereas its expression remained unaltered in embryos treated with control-MO (Ctrl-MO) and vector alone (Fig S3a). Similar reduction of Dlc1 expression was observed in Dlc1-MO-treated cultured NCCs, which appeared round in shape without distinct front-back polarity like DN-Dlc1 expressing cells (Fig. S3b and Fig. 3c). In agreement with this, cells expressing Dlc1-MO resulted in reduced expression of *FoxD3*⁺ migrating NCCs compared to the untransfected side and the Ctrl-MO-treated embryos (Fig. S3c). These results demonstrate functional requirements for Dlc1 in the establishment of NC polarity and delamination. Altogether these in vivo studies indicate that appropriate level of Dlc1 activity is required for the spatial restriction of RhoA activity, the establishment of NC polarity and directional delamination of NCCs. We have described the results on pages 11 and 12.

As suggested by the reviewer, we also evaluated RhoA activity and dynamics of migratory behavior in cultured neural crest cells overexpressing FRET biosensor with Dlc1 or DN-Dlc1 in the presence of SDF-1 (Fig. 4g). Consistent with in vivo observations, time-lapse imaging and the FRET index between the back and front revealed that overexpression and dominant-negative inhibition of Dlc1 activity resulted in marked reduction and elevation of RhoA activity respectively compared to the vector control, which exhibited a reproducible polarized distribution of RhoA activity with persistent high FRET signal at the back and migrated toward SDF-1 (Fig. 4h,i and Supplementary Videos 7-9). In addition, we observed a similar lack of cell polarity and directional movement in both treatments, suggesting that addition of SDF-1 in neural tube explant culture was not able to restore the morphological and migration defects caused by aberrant RhoA signaling (Fig. 4h). Altogether, these in vitro studies further consolidate in vivo results that delaminating NCCs require precise spatial activity of Dlc1 for restricting RhoA activation high at the back that is essential for the acquisition of polarized NCC morphology and directional migration toward the source of chemoattractant. We have described the results on pages 13 and 14.

We also measured RhoA activity in delaminating neural crest cells electroporated with FRET-biosensor and Nedd9-MO, Nedd9+DN-Dlc1 or Nedd9 (Fig. 6a,b) as well as in neural tube explants with the same treatment in the presence of SDF-1 (Fig. 6c-f). Quantification of the total FRET activity in cells treated with Nedd9-MO in vivo was comparable to the vector control and Nedd9 overexpressing cells (Fig. 6b), suggesting that loss of Nedd9 function and increased level of Nedd9 expression have no direct impact on total RhoA activity. However, we observed mislocalization of RhoA activity around the nucleus of Nedd9-MO-treated cell, which exhibited aberrant front-back polarization and lack of directionality in migration (Fig. 6a,c and Supplementary Video 10). Consistently the FRET index between the back and front of Nedd9-MO-treated cells did not exhibit differential character compared to the control and Nedd9 overexpressing cells (Fig. 6b,c,d). These data suggest that lack of RhoA polarity in the absence of Nedd9 function could be attributed to the dysregulated localization of Dlc1, which disrupted precise spatial RhoA restriction without altering the total level of RhoA activity. This prompted us to further examine whether association of Dlc1 with Nedd9 is functionally required for the establishment of RhoA polarity. We then overexpressed DN-Dlc1 to impair interaction of endogenous Dlc1 with Nedd9, rendering Dlc1 incapable for spatial restriction of RhoA activity at the cell rear. If that was the case, polarized RhoA activity could be restored by overexpression of Nedd9 in DN-Dlc1 expressing cells. Indeed, the differential distribution of RhoA activity between the back and front in DN-Dlc1+Nedd9 expressing NCCs in vivo was restored with the FRET index comparable to the control and Nedd9 overexpression alone, but its total FRET signal remained higher than other treatments (Fig. 6a,b). Clustering of the FRET index between the back and front of cultured NCCs expressing DN-Dlc1+Nedd9 also showed restoration of differential RhoA activity like Nedd9 overexpressing cells, and they exhibited polarized morphology and directional migration toward SDF-1 (Fig. 6c,e,f and Supplementary Videos 11 and 12). In agreement with this, restoration of NC delamination was observed in embryos electroporated

with DN-Dlc1+Nedd9 as opposed to a marked reduction of Sox10⁺ and HNK-1⁺ cells in the transfected side of embryos electroporated with DN-Dlc1 or Nedd9-MO, whereas overexpression of Nedd9 alone did not significantly affect NC delamination (Fig. 6g,h). Collectively, these data show that interaction of Dlc1 with Nedd9 is essential for the spatial restriction of RhoA activity at the back of the cell and the leading edge that is prerequisite for the establishment of a front-rear polarity axis and directional delamination and migration of trunk NCCs. We have described the results on pages 16-18.

We have analyzed substantial number of chick embryos, neural tube explants and neural crest cells for measurement of RhoA activity and migratory behavior analyses as indicated in the figure legends and highlighted a few examples below:

In Figure 1a-d, we have analyzed 25 chick embryos for quantification of the ratio of the FRET index between the back and front of delaminating (n=107), early migrating (n=123) and late migrating neural crest cells (n=196) on sections.

In Figure 1f-h, we have analyzed 81 delaminating neural crest cells from 26 neural tube explants for quantification of the total FRET index and the FRET index between the back and front. We have selected 12 representative neural crest cells from 12 explants (1 cell per explant) and used heat maps to represent their FRET index as a function of time at the back and at the front or their ratio of the FRET index between the back and front as a function of time based on the method published in Ramel et al., *Nat Cell Biol* 2013 15(3): 317-324. Similar representation was used to present FRET activity in cluster of neural crest cells for different conditions as shown in Fig. 6d-f.

In Figure 1i-k, we have analyzed 81 neural crest cells from 21 explants for quantification of the ratio of the FRET index between the back and front of polarized neural crest cells undergoing directional change of movement toward SDF-1.

In Figure 3c,d, we have quantified the total FRET index in 425 cells from 25 embryos treated with vector control, 131 cells from 16 embryos treated with Dlc1 and 148 cells from 21 embryos treated with DN-Dlc1.

2. Second, the authors present expression data of Dlc1 and claim this is localized to leading edge protrusions during delamination and migration. They present clear in vitro data (neural tube explant culture), but their in vivo data is both vague and under-represented (Fig. Sup 2). Why are there so few labeled neural crest cells and why is there no overlap in the Dlc1/GFP signal? Surely the GFP signal should be throughout the cell and not localized to the rear majority of the cell. Also, Fig Sup 2 shows Dlc1 is not present in all cell protrusions? Why did the authors not push the expression analysis in vivo?

Author Response:

Since chick-specific Dlc1 antibodies are not available and the one we used for immunofluorescence in explants did not work on sections, we generated a N-terminal V5-Dlc1 fusion construct in pCIG-IRES-nls-EGFP bicistronic vector to analyze the subcellular localization of Dlc1 in chick embryos. To avoid any saturation effect of Dlc1 overexpression and disruption of neural crest cell polarity, we overexpressed V5-Dlc1 fusion construct at a relatively low concentration (0.5µg/µl) that resulted in less number of neural crest cells being transfected. In Fig. S2, immunofluorescence staining on sections clearly shows that V5-Dlc1 (red) is preferentially localized in the cytoplasm to the front of delaminating and early migrating neural crest cells, which appear to follow their normal migratory route in ventral direction, and is expressed in a mutually exclusive with nuclear-localized EGFP (green). We did not find Dlc1 is present in all cell protrusions.

3. Third, manipulation of Dlc1 and subsequent affects on neural crest cell behaviors are measured in vitro. A stronger case would be to make these measurements in vitro in the presence of a guidance signal or in vivo, otherwise the overall speculation that Dlc1 regulates cell polarity and requirement for directional delamination is not conclusive. Furthermore, cells in which Dlc1 is over-expressed appear to be more amoeboid in shape but are not shown to migrate. This makes it difficult to evaluate RhoA signaling since the cells are not moving in any direction nor have significant protrusive activity. It is also difficult to argue that neural crest cells require precise spatial activity of Dlc1 for restricting RhoA activity low at the front since ctrl cells show high localized RhoA activity in active cell protrusions.

Author Response:

As mentioned above in response to 1, we studied the effects of overexpression of Dlc1 and DN-Dlc1 on RhoA activity; neural crest polarity and migration in both chick embryos (Fig. 3) and neural crest explant culture in the presence of SDF-1 (Fig. 4g-h). In chick embryos, cells overexpressing Dlc1 appear to be amoeboid in shape but still delaminated from the basal surface of the dorsal neural tube and into the lumen (Fig. 3c and e-h'). Similar lack of cell polarity and directional movement were observed in Dlc1 overexpressing neural crest explant culture in the presence of SDF-1 (Fig. 4h and Supplementary Video 8), suggesting that addition of SDF-1 in neural tube explant culture was not able to restore the morphological and migration defects caused by aberrant RhoA signaling (Fig. 4h). Altogether, these in vitro studies further consolidate in vivo results that delaminating neural crest cells require precise spatial activity of Dlc1 for restricting RhoA activation high at the back that is essential for the acquisition of polarized NCC morphology and motility toward the source of chemoattractant.

By performing FRET analysis in both chick embryos and time-lapse imaging of neural crest explants in the presence of SDF-1, we found in some occasion active RhoA is dynamically localized in cell protrusions (Fig. 1a, c, e) while we also observed cells without active RhoA in membrane protrusions of neural crest cells in vivo (Fig. 1b) and in cultured neural crest cells undergoing directional change (Fig. 1i and Supplementary Video 2) and front-back switch (Fig. 1l and Supplementary Video 3). In contrast, we frequently observed RhoA is consistently high in the cytoplasm at the cell rear in all conditions analyzed (Fig. 1). Quantification of the FRET index of delaminating and migrating neural crest cells clearly show high RhoA in the cytoplasm of the cell rear relative to low or fluctuating RhoA activity in membrane protrusion at the front (Fig. 1d,g,h). Since Dlc1 exhibits negative correlation with RhoA in subcellular localization (Fig. 2l,m), we have changed our conclusion statement on page 14 into "neural crest cells require precise spatial activity of Dlc1 for restricting RhoA activity high at the back."

4. *Fourth, it is difficult to judge whether Nedd9 interacts with Dlc1 since Nedd9 MO cells look more polarized than Ctrl-MO cells (Fig. 5g) and there is no quantitation of fluorescence as in all previous figures.*

Author Response:

We have analysed many neural crest cells treated with Nedd9-MO (n=112/12 neural crest explants; n=168/13 embryos) and the majority of them exhibited aberrant and elongated morphology without discernible front-back polarity (Fig. 5h and 6a). We also performed line scan analysis for the quantification of average intensity of Dlc1 expression in Nedd9-MO cells (Fig. 5i). The results show that Dlc1 protein exhibited random distribution throughout the Nedd9-MO transfected cells while polarized Dlc1 expression remained unaltered in Ctrl-MO-treated cells (Fig. 5h,i), suggesting that the asymmetric localization of Dlc1 is regulated by association with its binding partner Nedd9.

Reviewers' Comments:

Reviewer #1:

Remarks to the Author:

The authors have addressed my comments.

Although the single cell presented now in figure 1j supports the authors argument, I am reasonably confident that analysis of a larger number of cells would support the conclusion that asymmetry of RhoA activity sometimes indicates the future back-front polarity. I would therefore insist on inclusion of the word "sometimes" in the text on page 7.

Reviewer #2:

Remarks to the Author:

In their revised manuscript, the authors have addressed all my concerns.

Reviewer #3:

Remarks to the Author:

The authors have adequately addressed a large portion of my concerns. There is only one another point I suggest should be addressed and a small number of minor grammatical points.

1) Discussion section P22. The authors imply that their finding of the symmetric localization of RhoA activity is a key molecular signal that mediates neural crest cell orientation to the direction of movement, but was this not the similar finding of contact inhibition of locomotion and RhoA activity at the site of cell-cell collision leading to neural crest cell polarity as analyzed using a dynamic RhoA FRET activity reporter in Carmona-Fontaine et al., 2008. Furthermore, there is little discussion as to how the cell polarity axis changes and what are the FRET responses after cell-cell collisions in this study?

Minor points:

P2-Using a fluorescence resonance energy...

P2-...accompanied with the highly active...

P3-...at this stage are not known.

P5-that NCCs display differential RhoA activity..,

P7-preceded initiation of a gradual front and back switch...

P21- line spaces between paragraphs

Below please find our point-by-point response to reviewers' comments with changes highlighted in **red** in the revised manuscript (the reviewers' comments are in *italics*).

Reviewer #1 (Remarks to the Author):

The authors have addressed my comments.

Although the single cell presented now in figure 1j supports the authors argument, I am reasonably confident that analysis of a larger number of cells would support the conclusion that asymmetry of RhoA activity sometimes indicates the future back-front polarity. I would therefore insist on inclusion of the word "sometimes" in the text on page 7.

Author response:

We have included "sometimes" in our conclusion on page 7 as below: 'asymmetry of RhoA activity **sometimes** indicates the future back-front polarity and directional migration of trunk NCCs.'

--

Reviewer #2 (Remarks to the Author):

In their revised manuscript, the authors have addressed all my concerns.

--

Reviewer #3 (Remarks to the Author):

The authors have adequately addressed a large portion of my concerns. There is only one another point I suggest should be addressed and a small number of minor grammatical points.

1) Discussion section P22. The authors imply that their finding of the symmetric localization of RhoA activity is a key molecular signal that mediates neural crest cell orientation to the direction of movement, but was this not the similar finding of contact inhibition of locomotion and RhoA activity at the site of cell-cell collision leading to neural crest cell polarity as analyzed using a dynamic RhoA FRET activity reporter in Carmona-Fontaine et al., 2008. Furthermore, there is little discussion as to how the cell polarity axis changes and what are the FRET responses after cell-cell collisions in this study?

Minor points:

P2-Using a fluorescence resonance energy...

P2-...accompanied with the highly active...

P3-...at this stage are not known.

P5-that NCCs display differential RhoA activity..

P7-preceded initiation of a gradual front and back switch...

P21- line spaces between paragraphs

Author response:

We have included the following description in the discussion on pages 22 and 23:

In contrast, previous FRET studies in *Xenopus* embryos showed that RhoA activity is localized to the cell periphery and elevated at the site of collision of two cranial NCCs leading to contact inhibition of locomotion. NCCs arising from different axial levels may account for the differences in the subcellular localization of RhoA activity but its level of activity is similarly required for directional migration of both cranial and trunk NCCs. While cranial NCCs adopt contact inhibition of locomotion to govern their collective migratory behavior, trunk NCCs tend to migrate as single cell chains and interact extensively with neighboring cells. A recent report demonstrated that leader cells are crucial for orchestrating the orderly movement of trunk NCCs through cell-cell contact with follower cells. Based on these studies together with our results, it is tempting to speculate that leader cells may provide guidance signals through dynamic interaction with the followers to maintain differential RhoA activity and cell polarity axis for directed migratory patterns of trunk NCCs. Further analysis will be required to address this issue.

We have amended the minor points as requested except the following:

In the abstract on page 2, we have changed to “using the RhoA biosensor in vivo and in vitro” instead of using a fluorescence resonance energy...